# Comprehensive Review of Chronic Stress Pathways and the Efficacy of Behavioral Stress Reduction Programs (BSRPs) in Managing Diseases

**DOI:** 10.3390/ijerph21081077

**Published:** 2024-08-16

**Authors:** Aladdin Y. Shchaslyvyi, Svitlana V. Antonenko, Gennadiy D. Telegeev

**Affiliations:** Institute of Molecular Biology and Genetics, National Academy of Sciences of Ukraine, 150, Zabolotnogo Str., 03143 Kyiv, Ukraine; s.v.antonenko@imbg.org.ua (S.V.A.); g.d.telegeev@imbg.org.ua (G.D.T.)

**Keywords:** chronic stress, chronic psychological stress, stress reduction, behavioral stress reduction programs, BSRP, chronic psychological stress, diabetes, HIV, cancer, cardiovascular diseases

## Abstract

The connection between chronic psychological stress and the onset of various diseases, including diabetes, HIV, cancer, and cardiovascular conditions, is well documented. This review synthesizes current research on the neurological, immune, hormonal, and genetic pathways through which stress influences disease progression, affecting multiple body systems: nervous, immune, cardiovascular, respiratory, reproductive, musculoskeletal, and integumentary. Central to this review is an evaluation of 16 Behavioral Stress Reduction Programs (BSRPs) across over 200 studies, assessing their effectiveness in mitigating stress-related health outcomes. While our findings suggest that BSRPs have the potential to enhance the effectiveness of medical therapies and reverse disease progression, the variability in study designs, sample sizes, and methodologies raises questions about the generalizability and robustness of these results. Future research should focus on long-term, large-scale studies with rigorous methodologies to validate the effectiveness of BSRPs.

## 1. Introduction

Stress-related medical and psychiatric conditions are among the leading contributors to morbidity and mortality worldwide. These conditions include cardiovascular diseases, cancer, immune system disorders, post-traumatic stress disorder (PTSD), major depressive disorder, cognitive decline, psychotic disorders, and addictions. In Europe, mental disorders, primarily depressive and anxiety disorders, account for more than 60% of social and economic costs. Depression develops in response to everyday stressors, both major and minor. According to the World Health Organization, stress-induced chronic diseases are the leading cause of death in developed countries [1,2,3].

Stress significantly impacts health, not only by promoting disease processes but also by placing a substantial burden on healthcare systems. It plays a vital role in various modern ailments, particularly cardiovascular diseases, which are often aggravated by the psychosocial stressors of daily life, including work-related stress. In addition to pharmacological or clinical interventions, there is a critical need for effective behavioral stress reduction methods. Research has established a strong link between occupational stress and the risk of several cancers, including colorectal, lung, and esophageal cancers, in a study involving 281,290 participants. The adverse health effects of stress are extensive and diverse. Chronic stress induces substantial biological changes, such as increased apoptosis in the thymus and a reduction in thymocyte numbers. While there is an overall reduction in total lymphocyte count, not all types of lymphocytes are equally affected. The relative proportion of B cells experiences a modest decrease. Conversely, there is an increase in the relative proportion of CD3+ cells, particularly among T-cell subsets with an immature phenotype (CD3+PNA+). Neurobiological studies have identified the amygdala as a crucial brain region in stress responses, with research showing that reductions in perceived stress correlate with decreases in right basolateral amygdala gray matter density [4,5,6,7].

Gender differences significantly influence stress susceptibility and resilience. Women are more likely to develop autoimmune and affective disorders, while men have higher rates of early mortality, substance abuse, antisocial behavior, and infectious diseases. Poor stress management leads to severe physical and psychological consequences, affecting both individual and community health. The evidence indicates that adults in the U.S. predominantly use unhealthy strategies for managing stress [8,9].

Recent clinical trials support the hypothesis that the psychological or pharmacological inhibition of excessive adrenergic and inflammatory stress signaling, especially when combined with cancer treatments, can be life-saving. Between 1997 and 2003, a longitudinal study of chronic pain patients (n = 133) showed that those with arthritis, back or neck pain, or multiple comorbid pain conditions experienced significant improvements in pain intensity and functional limitations after participating in a Behavioral Stress Reduction Program (BSRP) [10,11].

Current medical therapies often do not achieve their desired effectiveness, imposing a substantial burden on global healthcare budgets. Extensive research indicates that stress can diminish the efficacy of medical treatments and exacerbate health conditions. Non-medical stress reduction interventions, particularly BSRPs, are increasingly recognized as promising approaches. Studies have shown that BSRPs have the potential to enhance the effectiveness of medical therapies and reverse disease progression. These programs have been demonstrated to improve various psychological and physiological parameters. This article evaluates the current landscape of non-medical stress reduction strategies, exploring their potential for widespread implementation to improve health outcomes and save lives.

Despite the enthusiasm surrounding BSRPs, one major concern is the variability in the design and methodology of studies assessing BSRPs. Many studies rely heavily on self-reported data, which can introduce significant biases. Additionally, the short duration of most studies limits the ability to draw conclusions about the long-term efficacy of these programs.

The terminology surrounding non-medical, non-pharmaceutical stress reduction programs is varied and often inconsistent. For the sake of clarity, we will use the term Behavioral Stress Reduction Program (BSRP) to refer to all such interventions throughout this review. Detailed descriptions of each BSRP will be provided in the Behavioral Stress Reduction Programs section.

## 2. Summary

To facilitate the understanding of this extensive review, we begin with a succinct summary of its core contents. Chronic stress exerts profound impacts on health, initiated when a stimulus is perceived by the mind as threatening. The brain is pivotal in this process, functioning as the primary organ that evaluates threats and coordinates both the behavioral and physiological responses.

Prolonged perception of a stimulus as a threat intensifies chronic stress, leading to more severe health consequences. The subsequent figure demonstrates the progression of stress-related consequences over time, as a stimulus continues to be perceived as a threat (Figure 1).

The accompanying table details specific genes triggered by chronic stress and their associated physiological outcomes, based on findings from various studies (Table 1).

Understanding that perception is a learnable skill, we can employ three strategies to manage stress:
Re-education involves transforming potential threats into beneficial opportunities through knowledge and understanding;Refocusing: shifting focus to stimuli already perceived as beneficial;Recoping: transitioning from harmful to healthy coping mechanisms.

These strategies are integral to transforming our stress responses. BSRPs leverage this key concept and have demonstrated significant outcomes. This review aggregates data from over 200 studies, offering detailed insights into the physical and psychological effects of BSRPs, accompanied by critical evaluations. The next figure provides a summarized schematic of the specific physiological benefits of BSRPs (Figure 2).

The comprehensive impact of stress on health and the mitigating effects of BSRPs are encapsulated in an overarching framework (Figure 3). This schematic illustrates the complex interplay between stress stimuli, physiological responses, and health outcomes, underscoring the crucial interventions provided by BSRPs to mitigate the adverse effects of chronic stress.

With this overview of the key findings, we now proceed to a deeper exploration of the detailed contents of this review.

## 3. How Stress Influences Health

The human body constantly reacts to internal and external stressors, processing these stimuli and eliciting responses based on perceived threat levels. The autonomic nervous system, comprising the sympathetic nervous system (SNS) and the parasympathetic nervous system (PNS), plays a crucial role in this process. Under stress, the SNS activates, triggering the fight-or-flight response through hormonal and physiological changes. The amygdala, responsible for processing fear and arousal, is key in this response, signaling the hypothalamus when necessary. Upon receiving a signal from the amygdala, the hypothalamus activates the SNS, prompting the adrenal glands to release catecholamines like epinephrine, which increase heart and respiratory rates. If the stress persists, the hypothalamus activates the hypothalamic–pituitary–adrenal (HPA) axis, leading to cortisol release from the adrenal cortex. Cortisol keeps the body on high alert by providing energy through its catabolic mechanisms [15,56].

The HPA axis, which regulates cortisol production and secretion, follows a circadian rhythm, with cortisol levels peaking in the morning and dipping at night. Cortisol, synthesized from cholesterol, serves multiple functions: mediating stress responses, regulating metabolism, and modulating the immune system. Its widespread influence extends to nearly every organ system, including nervous, immune, cardiovascular, respiratory, reproductive, musculoskeletal, and integumentary systems. In the immune system, glucocorticoids induce apoptosis in proinflammatory T-cells, suppress B cell antibody production, and reduce neutrophil migration during inflammation. Cortisol also regulates blood glucose levels by acting on the liver, muscles, adipose tissue, and pancreas. In the liver, it boosts gluconeogenesis and decreases glycogen synthesis, ensuring glucose availability for the brain. In muscles, cortisol reduces glucose uptake and increases protein degradation to supply gluconeogenic amino acids. In adipose tissue, it promotes lipolysis, releasing glycerol and fatty acids for energy. Additionally, cortisol reduces insulin secretion and increases glucagon levels in the pancreas, enhancing the effects of catecholamines and glucagon. Steroid hormones like cortisol can cross cell membranes and bind to cytoplasmic receptors, influencing gene transcription once inside the cell nucleus. For example, studies have shown that prenatal stress affects the expression of over 700 genes [56,57,58].

*FKBP5* is a critical modulator of stress responses, influencing glucocorticoid receptor activity and various cellular processes. Certain polymorphisms in the *DRD2* gene and the *SCL6A4* are linked to PTSD symptoms. These genetic variations can either increase or decrease sensitivity to stress and emotional disturbances. *NRXNs*, essential for neural circuit formation and remodeling, are also influenced by stress. Chronic psychological stress can alter Neurexin gene expression and splicing patterns, impacting synaptic strength regulation and potentially contributing to aversive conditioning. Polymorphisms in genes such as the *GR* gene affect HPA axis activity and sensitivity to stress hormones. The *5-HTT* and the *COMT* gene have also been linked to variations in stress-related traits and disorders, including neuroticism, anxiety, and depression [3,9,31,44,46].

Epigenetic changes, stable DNA modifications caused by environmental factors, can influence stress responses. For example, maternal care can affect genetic expression through DNA methylation, particularly in genes involved in glucocorticoid receptor expression in the hippocampus. Abused suicide victims show decreased levels of glucocorticoid receptor mRNA and an increased methylation of the receptor promoter compared to non-abused suicide victims [9,59].

The relaxation response (RR), which counters the stress response, can be triggered by BSRPs. Research indicates that both novice and long-term practitioners of RR experience significant gene expression changes, with long-term practitioners showing greater effects. RR enhances genes associated with energy metabolism, mitochondrial function, insulin secretion, and telomere maintenance, while reducing those linked to inflammation and stress [40,60,61].

### 3.1. Immune Response

Chronic stress significantly alters gene expression, impacting inflammation, immune response, and tissue development. Upregulated genes such as *Atg16l1* and *Coq10b*, and downregulated genes like *Abat* and *Cited2*, highlight the extensive biological changes induced by stress. A network enrichment analysis revealed that the most affected processes include the inflammatory response, chromatin remodeling, and immune cell signaling, while digestive system functions were notably downregulated. These genetic alterations underline the broad impact of chronic stress on physiological functions. In an animal model study, mice exposed to chronic unpredictable mild stress displayed behaviors indicative of depression and chronic stress. A differentially expressed genes (DEGs) analysis identified 282 DEGs, with the upregulated genes primarily involved in immune and inflammatory responses. The protein–protein interaction network highlighted ten hub genes related to the T-cell receptor signaling pathway, with increased expressions of *CD28*, *CD3e*, and *CD247* in stressed mice compared to controls. These findings suggest that immune pathways play a crucial role in the molecular mechanisms of chronic stress and may help identify potential biomarkers for early stress detection. Additionally, a study involving 207 participants examined the association between chronic stress and single nucleotide polymorphisms in genes related to immune response. The results showed that chronic stress is linked with higher levels of Th1 cells in individuals carrying specific minor alleles of the glucocorticoid receptor and interferon-gamma receptor genes. This indicates a genetic predisposition to stress-induced immune dysregulation [13,27,62].

### 3.2. Cancer

Chronic stress significantly impacts both brain function and cancer development through a cascade of harmful mechanisms. Stress hormones activated by the HPA axis and the SNS contribute to tumorigenesis and cancer progression by inducing DNA damage. Elevated glucocorticoid levels, for example, increase the activity of negative regulator murine double minute 2 (MDM2) via the induction of serum-and-glucocorticoid-regulated kinase (SGK1). This process inhibits p53, a protein crucial for DNA repair, thereby promoting tumorigenesis. Furthermore, chronic stress triggers an inflammatory response that fosters an inflammatory tumor microenvironment, facilitating all stages of tumorigenesis. It also enhances neuroinflammation, impairing the brain’s ability to process stress. Chronic stress selectively suppresses type 1 helper T-cells (Th1) and cytotoxic T-cell (CTL)-mediated cellular immunity, reducing immune surveillance and increasing the risk of cancer invasion and metastasis. This suppression compromises the effectiveness of anti-tumor therapies [63,64].

Chronic stress disrupts the critical immune cell functions involved in anti-tumor immunity, inhibiting dendritic cells and lymphocytes, while enhancing the activity of tumor-supportive cells such as myeloid-derived suppressor cells and tumor-associated macrophages [65].

The stress response gene *SIRT1*, activated by *BCR-ABL*, promotes leukemogenesis by enhancing chronic myeloid leukemia (CML) cell survival and proliferation. *SIRT1* activation involves the deacetylation of multiple substrates, including FOXO1, p53, and Ku70. Additionally, variations in genes related to oxidative stress, such as CAT-262 C>T, *GPX1* Pro198Leu, and *GSTP1* Ile105Val, influence the risk of developing *BCR-ABL* negative myeloproliferative neoplasms [18,19,20].

### 3.3. Gastrointestinal Illnesses

Stress influences appetite regulation by affecting the central appetite/satiety centers in the hypothalamus. Corticotropin-releasing hormone (CRH) can acutely induce anorexia, while NPY stimulates CRH secretion via Y1 receptors, counter-regulating its own effects. Additionally, NPY inhibits the sympathetic system and activates the parasympathetic system, reducing thermogenesis and facilitating nutrient digestion and storage [27,66].

Historical events, such as the 1957 Delaware River tanker collision, illustrate the profound psychological and somatic effects of acute stress. Survivors experienced immediate psychological symptoms like nervousness and anxiety, along with GI complaints. Long-term follow-up revealed a significant psychological deterioration, with many survivors reporting headaches, restlessness, phobic reactions, depression, sleep disturbances, and severe GI illnesses [67].

### 3.4. Diabetes

The serotonin 5HTR2C receptor plays a crucial role in mediating HPA axis activation during stress, which can have broader implications for other stress-related conditions such as diabetes. A study investigating the functional polymorphism (rs6318) of the *5HTR2C* gene in 41 men found that individuals with the more active Ser23 C allele exhibited significantly higher cortisol levels and greater increases in anger and depressive mood during laboratory stress compared to those with the less active Cys23 G allele. These findings suggest that genetic variation in *5HTR2C* is associated with HPA axis activation and emotional stress responses, which may increase the risk of cardiovascular disease and type-2 diabetes, linking the mechanisms of stress with multiple health outcomes [12].

### 3.5. Epilepsy

Epilepsy and depression often co-occur, particularly in cases of temporal lobe epilepsy, linked by dysfunction in the HPA axis. Chronic stress leads to excessive glucocorticoid production, which disrupts hippocampal function—a brain region critical for learning, memory, and emotional regulation. The hippocampus’s high functional plasticity makes it particularly vulnerable to stress-induced damage, mediated by glucocorticoid hormones. This results in neurodegeneration, loss of hippocampal neurons, and altered neurogenesis, contributing to the development of epilepsy and depressive disorders through shared molecular pathways involving neurotransmitters and neurotrophic factors [68].

### 3.6. Parkinson’s Disease

Parkinson’s disease is another condition where chronic stress plays a detrimental role. Chronic stress exacerbates inflammatory responses in midbrain microglia, suggesting a potential risk factor in the degenerative processes characteristic of Parkinson’s disease. This sensitization may accelerate disease progression and worsen symptoms, highlighting the importance of stress management in these patients. Effective stress management could potentially slow disease progression and enhance patients’ quality of life [69].

### 3.7. Depression and Anxiety

Recurrent depressive disorders have emerged as a significant problem, with biological theories increasingly focusing on active inflammatory processes within the body. Inflammatory markers such as manganese superoxide dismutase (MnSOD), myeloperoxidase (MPO), and inducible nitric oxide synthase, along with proinflammatory and anti-inflammatory cytokines, play crucial roles in the pathogenesis of depression. The kynurenine pathway, which leads to deficits in serotonin and melatonin, is also implicated. Chronic stress, acting through these pathways, can be seen as a catalyst for depression, likened to a chronic cold of the organism responding to everyday stressors. Chronic stress exposure significantly increases the risk of developing various neuropsychiatric illnesses. For example, chronic restraint stress (CRS) has been shown to cause dendritic hypertrophy and glutamate-related synaptic remodeling in basolateral amygdala projection neurons (BLA PNs), particularly those targeting the ventral hippocampus, correlating with anxiety-like behaviors. Similarly, patients with somatic symptom disorder (SSD) who underwent BSRP showed significant reductions in depression, anxiety, stress, and physical symptoms, compared to controls [1,70,71].

Furthermore, genetic predispositions also play a critical role in the stress response and mental health disorders. For example, carrying the short allele of the *5-HTTLPR* in high-stress environments increases depression risk. Effective emotion regulation can mitigate these effects, highlighting the importance of learnable skills in resilience. Polymorphisms in genes such as *EPHX2*, *OXTR*, and *NRG1* are associated with depression symptoms, further underscoring the complex interplay between genetics and environment. An altered expression of the *PPM1F* gene in the amygdala and medial prefrontal cortex (mPFC) is linked to PTSD and depression. Similarly, polymorphisms in the *5-HTTLPR* gene correlate with cognitive dysfunction in PTSD-affected individuals [33,45,48,49,50].

### 3.8. PTSD

PTSD is another severe mental health condition exacerbated by stress. Gene expression analysis in chronic PTSD patients has identified significant dysregulation in the immune, endocrine, and nervous pathways. Key genes, such as *PRKCA*, *TP53*, *EP300*, and *CALM1*, are implicated in the pathogenesis of PTSD, providing insights into the molecular basis of the disorder. Early life adverse social conditions can alter gene expression, leading to increased proinflammatory and reduced antiviral gene activity. Studies on former child soldiers show that high personal resilience can buffer these effects, reducing PTSD severity and normalizing gene expression profiles. The *Wdr13* gene is upregulated in the hippocampus of mice exposed to stress, leading to significant behavioral and molecular changes. Social isolation in *Wdr13*-deficient mice results in heightened anxiety, anhedonia, and reduced dendritic branching. Molecular analysis reveals the downregulation of synaptic proteins and upregulation of *GATA1*, a transcription factor linked to major depression in humans [26,36,72].

Further research into PTSD mechanisms has revealed that psychosocial stress alters the expression of the acetylcholinesterase gene, generating proteins implicated in various pathologies, including PTSD. Genetic studies indicate that *NPSR1* and *GAD1* polymorphisms may contribute to PTSD development among individuals affected by the Balkan wars in the 1990s. Studies using rat models of PTSD have identified significant temporal differences in gene expression patterns, highlighting pathways such as *EIF2*, *NRF2*-mediated oxidative stress response and sirtuin signaling [32,35,73].

### 3.9. Accelerated Aging, Early Mortality

Chronic stress is a significant contributor to accelerated aging, primarily through mechanisms involving inflammation and oxidative stress. Inflammation, often termed “inflammaging”, is a chronic, low-grade inflammatory state that is closely associated with aging. It contributes to various age-related pathologies, including atherosclerosis, diabetes, and hypertension. Oxidative stress, caused by the overproduction of reactive oxygen species (ROS), further exacerbates tissue damage. Mitochondria are the primary source of ROS, and their dysfunction due to suppressed mitophagy amplifies pro-inflammatory changes specific to aging. Macrophages, key players in innate immunity, are central to chronic inflammation and inflammaging [74].

Experimental studies provide insights into the cellular and molecular alterations induced by chronic stress. In a study involving male SCID mice subjected to 14 days of restraint stress, researchers observed an increased expression of aging markers such as p16INK4a and p21, alongside reduced DNA damage repair capabilities. These stressed mice also exhibited a higher expression of *SASP*, *CREB*, *NF-κB*, and *AP-1* genes. The findings suggest that chronic stress can significantly affect biological aging pathways, particularly within bone marrow leukocytes, potentially mediated by sympathetic beta-adrenergic receptor activation. The role of *USP1* in endoplasmic reticulum stress and DNA replication stress further underscores the complexity of stress-related cellular aging [16,52,53,54,55].

Telomeres, the protective caps at the ends of chromosomes, are highly sensitive to chronic stress. A shortened telomere length is linked to increased cancer incidence and mortality, and populations under chronic stress exhibit accelerated telomere shortening. BSRPs have shown promise in mitigating this effect. In one study, BSRPs were significantly associated with increased telomere length in CD14(+) monocytes, with similar trends observed in CD14(-) T lymphocytes. Longitudinal increases in the naive T-cell population were also noted, suggesting a rejuvenating effect on immune cells. Furthermore, in women with breast cancer, telomerase activity—a key enzyme in maintaining telomere length—increased by approximately 17% over 12 weeks in the BSRP group, compared to negligible changes in the control group [75,76].

Social isolation and loneliness are additional stressors that exacerbate aging-related inflammation. Lonely older adults display an increased expression of pro-inflammatory genes and higher risks for morbidity and mortality. A study involving older adults demonstrated that BSRPs effectively reduced loneliness and downregulated pro-inflammatory *NF-κB*-related gene expression in circulating leukocytes [77].

### 3.10. Unhealthy Coping Responses to Stress

Coping mechanisms are the conscious thoughts and behaviors mobilized to manage internal and external stressful situations. Unlike defense mechanisms, which are subconscious, coping mechanisms are deliberate actions aimed at reducing or tolerating stress. Healthy coping strategies can be categorized into problem-focused, emotion-focused, meaning-focused, and social coping. Problem-focused strategies address the source of stress directly, such as active coping and planning. Emotion-focused strategies aim to manage the emotional distress associated with stress through positive reframing and acceptance. Meaning-focused strategies involve deriving meaning from stressful situations, while social coping involves seeking emotional or instrumental support from others. In contrast, maladaptive coping mechanisms, such as disengagement, avoidance, and emotional suppression, are associated with poor mental health outcomes and increased engagement in health risk behaviors, including smoking and alcohol consumption [78].

#### 3.10.1. Smoking and Alcohol Consumption

Stressful events can trigger affective responses like worry and anxiety, altering the functions of the HPA axis and the sympathetic–adrenal–medullary (SAM) system. This can lead to poor health decisions, such as decreased exercise, poor sleep, increased smoking, and alcohol consumption, which ultimately increase disease risk. The pathway from stress to disease involves environmental demands, perceived stress, negative emotional responses, poor health behaviors, and physiological changes, culminating in a heightened risk of disease onset or progression. Smokers often relapse after cessation programs, citing stress as a major factor. A study of over 136,000 citizens showed a significant correlation between high stress levels and increased tobacco consumption [79,80,81,82].

#### 3.10.2. Drugs and Hypersexual Behavior

Stress and the neurobiology of the stress response play critical roles in substance use initiation, maintenance, and relapse. Stress mediators interact with the dopaminergic reward system and other systems involved in addiction, such as endogenous opioids, the SAM system, and endocannabinoids. Emerging research highlights the role of stress in behavioral addictions, including hypersexual behavior, gambling, dysfunctional internet use, and food addiction. For instance, HIV transmission risk is notably high among men who have sex with men and women (MSMW), especially those with histories of childhood sexual abuse and current traumatic stress or depression. BSRPs have been shown to be more effective than general health promotion interventions in reducing risky sexual behaviors and depression symptoms in this population. Additionally, adolescence is a critical period for the maturation of behavior, dopamine systems, and the HPA axis. Stress during adolescence can increase sensitivity to drugs of abuse and alter the final maturation of cortical dopamine, potentially through mechanisms involving D2 dopamine receptor regulation and the glucocorticoid-facilitated pruning of dopamine fibers [83,84,85].

Drugs of abuse, such as opioids, psychostimulants, and alcohol, can induce neuroinflammatory mediators like interleukin-1β (IL-1β), which modulate the drug reward, dependence, and tolerance. These substances may directly activate microglial and astroglial cells via Toll-like receptors (TLRs), contributing to neuroinflammation and the addictive process. Stress is also a significant risk factor for the initiation and maintenance of cannabis use [86,87].

#### 3.10.3. Gambling

Individuals with gambling disorder exhibit significantly higher rates of post-traumatic stress symptoms (PTSS) and PTSD compared to the general population. Analysis of data from the National Epidemiologic Survey of Alcohol and Related Conditions, which included 41,869 participants, revealed that those identified as at-risk gamblers and problem gamblers were more likely to have multiple Axis-I and Axis-II disorders than non-gamblers, with this relationship being stronger in low-stress environments. The sensitization of the mesolimbic dopamine pathway, observed in both gambling and substance use disorders, underscores the significant role of stress in relapse [88,89,90].

#### 3.10.4. Compulsive Buying–Shopping Disorder (CBSD)

A scoping review encompassing 16 studies identified a strong correlation between general perceived stress and CBSD symptom severity. Individuals with problematic buying–shopping behavior demonstrated higher levels of perceived stress compared to controls. An online shopping addiction tendency (OSAT) among college students is a growing concern, particularly in China. A study involving 1123 students from eight universities in Guangdong Province found a significant relationship between stress (academic, personal, and negative life events) and OSAT scores [91,92].

#### 3.10.5. Overeating

Chronic stress and emotional states significantly impact eating behaviors. Positive emotions such as interest and surprise were linked to low-fat eating, while negative emotions like anger, shyness, and guilt were associated with emotional eating. A study of Chinese college students during the COVID-19 pandemic showed that stress and anxiety were indirectly linked to overeating. Research on binge eating disorder highlights the strong link between stress and binge eating, with state-level stress and its negative affects correlating with binge eating episodes in both laboratory and natural settings [93,94,95,96].

#### 3.10.6. Media and Games

Stress during the COVID-19 pandemic led to an increased use of social media, television, YouTube, and streaming services. A survey of 685 respondents found that those who reported their mental health as “not good” were more likely to use streaming services as a coping mechanism. Increased stress was linked to more hedonic and escapist media use, suggesting that media consumption can become a maladaptive coping strategy. A cross-sectional study of 400 community dwellers in suburban Delhi revealed that many individuals used mobile phones, television, and social networking sites as stress busters, highlighting the potential for media to serve as both a stress reliever and stressor. Combining this data with earlier findings, where media and social networks frequently deliver stressful content, reveals a perilous cycle of escalating stress. Individuals may initially turn to these platforms seeking relief from stress and immediate gratification. However, over time, this consumption of news and media can exacerbate their stress levels, creating a feedback loop of increasing stress. This phenomenon, perpetuated on a massive scale, affects hundreds of millions globally through the pervasive reach of mass media [97,98,99].

#### 3.10.7. Violence

Nursing professionals face extraordinary stressors in the medical environment, including long work hours, emotional suffering, and complex patient care. This stress has been linked to high incidences of depression and chronic disease among nurses. Additionally, a stressful work environment can lead to lateral violence, including behaviors such as verbal affronts, scapegoating, sabotage, and backstabbing. A study involving 370 inpatients at the Center for Inpatient Forensic Therapy in Zurich found that a higher number of stressors correlated with more violent offenses [100,101].

#### 3.10.8. Suicide

Suicide is a global health issue, with at least 800,000 deaths annually. Many models explain suicide risk, with a central component being the interaction between acute stress and susceptibility to suicidal behavior. Dysregulated HPA axis activity, as measured by cortisol levels, is a significant risk factor for suicide. Data from over 16,000 adolescents revealed an increasing trend in psychosocial stress, loneliness, and suicide ideation from 2006 to 2016, with violence being a significant contributor to stress for both boys and girls. A cohort study of 214,649 men with prostate cancer found that those with preexisting PTSD had higher suicide rates, emphasizing the critical role of stress disorders in suicidal behavior. A study of 780 students showed a positive correlation between childhood trauma, stress, sleep disturbances, and suicidal behavior. Research from Denmark indicated that stress disorders substantially increased the rate of suicide attempts, with significant associations observed for various types of stress disorders [102,103,104,105,106].

### 3.11. Conclusions

In summary, stress profoundly impacts health by altering physiological systems and gene expression. Chronic stress activates the HPA axis, leading to sustained cortisol release, resulting in immune dysregulation, inflammation, and neuroplasticity changes.

This stress is associated with higher risks of cancer, gastrointestinal issues, diabetes, neurological conditions, and mental health disorders through genetic and epigenetic mechanisms. Additionally, it accelerates aging via inflammation and oxidative stress while also promoting maladaptive coping behaviors such as substance abuse, compulsive behaviors, overeating, excessive media consumption, violence, and suicidal tendencies.

## 4. Sources of Chronic Stress

### 4.1. Work and Education

Chronic stress in work and educational environments significantly impacts mental and physical health. The findings revealed that occupational stress significantly impacts mental health, with genetic interactions involving *BDNF* rs10835210 and *TPH2* variants further increasing the risk of mental disorders. Medical students under examination stress exhibit reduced IL-2R-positive cells and IL-2R messenger RNA in blood leukocytes, suggesting a compromised immune function [21,107,108].

Stress resilience, the ability to manage anxiety under stress, is influenced by genetic factors. Variants in genes like *NPY*, *CRHR1*, and *5-HTT* are linked to lower resilience, while upregulation of *DRD1*–*DRD4*, *DBH*, *DAT*, and *BDNF* is associated with higher resilience. Research on medical students facing license examinations revealed significant changes in gene expression profiles in peripheral blood, validating changes in 10 key genes (*DPYD*, *CSF3R*, *PLCB2*, *CTNNB1*, *PPP3CA*, *IRF3*, *POLM*, *CCNI*, *ARHGEF1*, and *TP53*). Similarly, studies on law students found that genetic variability in the *NPS*/*NPSR1* system becomes more evident under chronic stress conditions like exam preparation [17,22,41,109].

Interpersonal stress and inadequate sleep can also activate *NF-κB* signaling pathways, leading to increased inflammation, especially during late adolescence. The combination of these stressors significantly enhances *NF-κB* activation [110].

### 4.2. Old Age

Old age presents unique challenges that significantly impact physical, emotional, and psychological well-being. Depression is the most common mental health issue among older adults, contributing to increased medical morbidity and mortality, reduced quality of life, and elevated healthcare costs. Early diagnosis and effective management are crucial to improving the quality of life for older adults with depression [111].

### 4.3. Prenatal Stress

Prenatal stress has profound implications for the development and health of offspring, influencing a wide range of physiological and genetic outcomes. Research has identified over 700 genes in the frontal cortex and hippocampus that are differentially expressed following prenatal stress. Adverse experiences during the perinatal period have been linked to methylation changes in the *NR3C1*, which are associated with long-term diseases. Hypermethylation under stressful conditions suggests a mechanism through which prenatal stress can program future health risks [23,57].

In a study involving pregnant Sprague Dawley rats, daily stress during the last week of gestation led to significant alterations in the offspring’s HPA axis and immune competence. For example, the corticosterone response to immobilization stress disappeared in control animals after 10 days but persisted in prenatally stressed animals. Additionally, prenatal stress decreased the total peripheral leukocyte count, altered differential counts by decreasing lymphocytes and increasing neutrophils and eosinophils, and significantly reduced the percentage of the peripheral lymphocyte T CD8+ subset in male offspring. These findings highlight the potential long-term impacts of prenatal stress on immune function and stress responsiveness. Also, in mice, exposure to predator odors during pregnancy increased corticosterone response and altered gene expression related to stress in the offspring. Specifically, there was an elevated transcript abundance of the *CRHR1* in the amygdala and decreased the *BDNF* transcript abundance in the hippocampus, correlated with site-specific DNA methylation changes [39,112].

Maternal stress can also influence pregnancy outcomes and offspring health in a gender-specific manner. For instance, maternal stress increased gestational length variation, adverse pregnancy outcomes, and reduced gestational weight gain. Prenatal stressors negatively affected pup weight, with differential impacts based on gender. Moreover, stress upregulated neuroendocrine and cytokine gene expression in dams, while offspring exhibited varied transcriptional responses depending on the specific stressors experienced [30].

### 4.4. Traumatic Events

Traumatic events are a pervasive source of stress, affecting a significant portion of the population. In North America, it is estimated that 60% to 75% of individuals experience a traumatic event at some point in their lifetime. These events include sexual assault, repeated childhood abuse or neglect, serious accidents, and exposure to war. Women who have experienced early life abuse are particularly vulnerable, as such trauma initiates pathophysiological cascades leading to long-term maladaptive stress responses, hyperalgesia, and an increased risk of psychopathology [113,114].

Interpersonal trauma can lead to prolonged physiological imbalances, increased rates of illness, and early mortality. Bereavement stress significantly increases mortality rates, with studies showing that mortality rates for men rise by 40% in the first six months following the loss of a spouse, and close relatives of the deceased have a 7.9 times higher mortality rate within a year compared to a control group [115,116].

Economic difficulties also contribute to stress, with a low income being associated with higher concentrations of interleukin-6 across all racial and ethnic groups. Persistent economic difficulties predict increased sickness absence, highlighting the broad impact of financial stress on health [117,118].

Animal studies provide valuable insights into the biological impact of stress. Chronic stress in dairy cows, induced by overcrowding and exposure to stressful noises, led to decreased milk production, altered activity, and increased physiological stress markers. Similarly, bovine respiratory disease complex in beef cattle is triggered by physical, biological, and psychological stressors, illustrating the economic and health impacts of stress in livestock [119,120].

In laboratory settings, environmental stressors such as social defeat and isolation significantly alter gene expression in the brain. For instance, chronic isolation in female rats reduced the *DRD1* gene expression in the amygdala, while increasing dopamine concentration in the VTA. Social defeat stress in male rats affected pituitary gland gene expression related to neuron morphogenesis and communication, with genetic variability in the *NRCAM* gene influencing stress responses in humans exposed to abusive supervision [43,121].

Studies on the genetic mechanisms underlying stress responses show that positive emotions can neutralize the moderating effects of the *BDNF*(Val66Met) genotype on social stress sensitivity, indicating a potential avenue for mitigating stress through emotional regulation. In humans, early life stress has been significantly associated with DNA methylation levels at the *5-HTTLPR* site, correlating with decreased white matter integrity in individuals with panic disorder. The importance of gene regulation in response to stress is further underscored by studies examining the roles of cis- and trans-regulatory factors in the prefrontal cortex and amygdala. Chronic social defeat (CSD) in mice reveals that while cis-regulatory mechanisms predominantly control gene expression in these brain regions, trans-regulatory mechanisms become more prominent in the amygdala under stress. This shift highlights the complex, site-specific regulatory patterns that psychological triggers can induce [24,47,122].

Acute stressors, such as exposure to a natural predator, significantly increase fearful and anxious behaviors in rats, accompanied by a robust activation of the amygdala and increased expression of somatostatin receptor 2 (sst2) mRNA. Similarly, chronic restraint stress (CRS) in mice results in higher expression levels of genes like *Cldn2* and *Nr1h4*, reflecting widespread changes in gene expression associated with sustained stress. Stress also impacts appetite regulation through complex neurobiological pathways. A study on the long-term effects of a single exposure to immobilization stress shows that stress-induced changes in anorexigenic and orexigenic factors in the hypothalamus and dorsal vagal complex (DVC) can lead to compensatory mechanisms that abolish stress-induced anorexia. This finding underscores the intricate balance between different neuropeptides in maintaining homeostasis under stress [28,42,123].

Early emotional trauma has profound and long-lasting impacts on neurobiological development. Childhood abuse and neglect deregulate the developing neurobiological system, reducing resistance to stress and leading to later problems with emotional regulation. These adverse experiences alter brain structures, such as increased synapse formation and dendritic growth in the basolateral amygdala, and dendritic retraction in the hippocampus, contributing to anxiety-like behaviors. The*NR3C1* is particularly susceptible to DNA methylation modifications in response to early life trauma, increasing the risk of depression later in life [37,124].

### 4.5. Parenting, Caring, and Nursing

Stress is an intrinsic part of many caregiving roles, whether in parenting, nursing, or providing long-term care for patients with chronic illnesses. These roles present unique challenges that contribute to high stress levels, impacting both physical and mental health. For instance, caregivers of patients with Alzheimer’s disease experience significant psychological stress, which negatively affects their immune response. A study on the antibody response to the tetanus vaccine in these caregivers found that higher baseline stress and depression scores were inversely associated with the antibody fold increase post-vaccination, underscoring the broader health implications of caregiving stress [125].

Nurses, especially those working in intensive care units and caring for dying patients, face considerable job-related stress. Psychiatric nurses report even higher work stress and lower mental health levels compared to their peers in other nursing specialties. The intense emotional demands and constant exposure to patient suffering contribute to chronic stress and burnout, emphasizing the need for effective stress management strategies [126,127].

### 4.6. Media

Media consumption, especially during crises, profoundly impacts psychological and physiological well-being due to its extensive reach and rapid dissemination of information. Unlike other personal stressors discussed in previous sections, the media stands out as a potent and pervasive source of stress, making it crucial to examine its effects. Excessive media consumption has been linked to heightened stress and negative emotional states, with individuals who consume media for over four hours daily reporting higher Perceived Stress Scale scores and lower Positive Emotion Scale scores compared to those with less media exposure. Moreover, problematic social media use (PSMU) has been associated with severe depression, anxiety, and stress symptoms [128,129,130].

The impact of traumatic media exposure is particularly profound. For instance, a study of New York City employees three years after the 9/11 attacks revealed that early media contact and distress from graphic images were linked to re-experiencing symptoms and hyperarousal, even among unexposed participants. Similarly, extensive media coverage of the Paris terrorist attacks correlated with higher PTSS, especially in those exposed to over four hours of daily media. In Chinese populations, prolonged media exposure and social media use were associated with higher acute stress and probable acute stress disorder. Another study involving 404 socioeconomically diverse adults found that higher frequencies of TV viewing, media sharing, and online friendships were significantly associated with greater PTSD symptom severity, even after controlling for depression severity [131,132,133,134].

The effects of media coverage on stress and trauma are further illustrated by events such as the December 2004 tsunami and Typhoon Hato. In Hong Kong, frequent exposure to distressing tsunami-related news was significantly associated with post-traumatic stress symptoms, while in Macao, indirect exposure to disaster-related social media content following Typhoon Hato was linked to PTSD among university students. These findings underscore the powerful role the media plays in amplifying stress and trauma in affected populations [135,136].

The role of the media in propagating stress is also highlighted by studies showing that increased television viewing and social media use following traumatic events can elevate PTSD prevalence in communities. Microsimulation models suggest that reducing media exposure could significantly decrease PTSD prevalence. Additionally, studies indicate that social media use exacerbates the stress symptoms and burdens caused by COVID-19, highlighting the complex interplay between media consumption and stress [137,138,139].

Interestingly, while social media can exacerbate stress, it can also provide support and connection, thus playing a dual role. High levels of doomscrolling—the obsessive consumption of negative news—are associated with decreased mindfulness and increased secondary traumatic stress, while social media can alleviate stress by providing factual and positive information. This support is crucial in mitigating the effects of sensational and false news. Dr. Champion Kurt Teutsch advocates for a “diet of good news” to help restore mental and physical health, emphasizing the importance of mindful media consumption. In this context, it is evident that while the media can exacerbate stress, it also plays a role in coping when used appropriately. A study involving 685 participants indicated that the media, especially social networks, can provide support and connection through the dissemination of factual and positive information. This support is crucial in mitigating the overflow of sensational and false news, which can otherwise increase stress levels. Therefore, the dual nature of social media highlights the importance of a balanced consumption and the potential benefits of positive and factual content in reducing stress [98,140,141,142,143,144].

### 4.7. Conclusions

In summary, stress from work, education, old age, prenatal conditions, traumatic events, caregiving roles, and media consumption leads to immune dysregulation, inflammation, and altered gene expression. Traumatic events and economic difficulties trigger maladaptive stress responses, increasing illness rates and mortality. Excessive media consumption, especially during crises, exacerbates stress. The following sections will explore how BSRPs can mitigate these detrimental effects and promote health.

## 5. The Influence of BSRPs on Health

### 5.1. Perception: The Central Pillar in Stress

While much focus is directed towards managing stress and its consequences, there is a critical aspect that often lacks global attention: the perception of stress. Stimuli are perceived by our minds as threatening. The brain plays a pivotal role in perceiving and responding to stress, acting as the primary organ that assesses threats and orchestrates both behavioral and physiological responses. This dual role facilitates adaptation, a process known as allostasis, but can also lead to pathological conditions when the system is overburdened, resulting in allostatic load or overload. The brain’s remarkable structural and functional plasticity allows it to adapt to stressful and diverse experiences through mechanisms such as neuronal replacement, dendritic remodeling, and synapse turnover. Stress can disrupt the neural circuits that govern cognition, decision-making, anxiety, and mood, leading to alterations in behavior and emotional states. This neural imbalance influences systemic physiology through neuroendocrine, autonomic, immune, and metabolic pathways. Initially, these physiological changes may be adaptive, helping the body to cope with immediate threats. However, if the stressor persists and the maladaptive state continues, these changes can become detrimental. Chronic stress-induced neural alterations may necessitate intervention, combining pharmacological treatments and behavioral therapies to restore balance and promote recovery [145,146,147].

### 5.2. Perception Is a Learnable Skill

By gaining knowledge and learning how to reinterpret stimuli, we can transform our responses to stress. Our ancestors learned to use previously threatening stimuli to their advantage: from turning wolves into guardians, to harnessing the sun’s energy for electricity. This ability to learn and adapt opens new possibilities for managing stress by addressing perception, potentially freeing us from the cascade of stress-related consequences. For example, genetic predispositions, such as carrying a short allele in the *5-HTTLPR*, are linked to an increased risk of depression in stressful environments. However, the capacity to regulate emotions effectively can mitigate this risk. In a study of 205 children aged 9–15, those who were genetically predisposed to higher stress levels exhibited more depressive symptoms. Crucially, at-risk children who practiced effective emotion regulation did not show heightened depressive symptoms, demonstrating that emotion regulation is a powerful tool for resilience. Perception also plays a significant role in how stress is experienced and managed in other contexts. For instance, a study involving 265 parents of children with hematological cancer or solid tumors revealed that parents’ perceptions of their child’s pain and their own stress levels were influenced by their attitudes towards analgesics. Parents with a lower educational attainment reported higher stress and more negative views on pain management, due to misconceptions about analgesic use. This underscores the importance of education and perception in managing stress and pain effectively [48,80,146,148,149,150].

### 5.3. Over 200 BSRP Studies

By leveraging the key concept that perception is a learnable skill, BSRPs have amassed a significant number of outcomes over time. In this review, we have collected over 200 studies detailing the physical and psychological consequences of BSRPs, along with critical evaluations. The most efficient and convenient way to present these studies is in a standalone table (Table 2).

### 5.4. Conclusions

By compiling the data presented, we can categorize how BSRPs address stress into three main strategies:
Re-education;Refocusing;Recoping.

Re-education involves transforming potential threats into beneficial opportunities through knowledge and understanding. This approach emphasizes shifting perceptions from being a victim to becoming a beneficiary. Here is a highly simplified example: understanding the physics behind lightning can turn fear into safety—lightning seeks the shortest path from clouds to the ground, so equipping a house with a lightning rod in advance allows one to relax during a storm and safely enjoy the enhanced ozone in the air, transforming a perceived threat into a moment of pleasure. Regularly increasing conscious awareness is crucial for recognizing and addressing cognitive biases. Many stimuli perceived as threatening are processed subconsciously, making them difficult to manage. By enhancing their awareness, individuals can identify these biases and eliminate them through re-education. This method aims to clear the mind of perceived threats that are, in reality, cognitive biases, by teaching how to use stimuli to one’s advantage. Re-education is an ongoing process of learning and understanding that is fundamental to all branches of science.

Refocusing is a strategy that involves redirecting attention to stimuli that promote relaxation. This can be achieved through various methods, including engaging with calming images, videos, films, affirmations, mantras, music, pets, or nature. Practices such as meditation and mindfulness are particularly effective, as they facilitate a shift from automated subconscious processes to conscious awareness. By focusing on the present moment, individuals can perceive it as a neutral stimulus, free of judgment, and consciously change their perception towards curiosity, openness, and acceptance. Concentrating the imagination to visualize tranquil scenes can also be beneficial. Another aspect of refocusing is “reframing”, which involves putting a positive spin on situations. This means finding something good in a bad situation and focusing attention on this positive aspect. While reframing can be helpful in mitigating stress, it is important to recognize that all forms of refocusing are temporary solutions. When the stressor inevitably returns, or when confrontation is unavoidable, relying solely on refocusing can become problematic. For example, if an individual lives on top of a mountain where lightning frequently strikes, ignoring the imminent threat by focusing on beautiful butterflies and sunlight can lead to severe consequences. Although stress might be avoided temporarily, serious harm remains a risk. In such scenarios, refocusing can serve as a temporary measure to regain composure and energy. This short-term strategy can provide the necessary mental space to refocus and motivate engagement in more effective long-term solutions, such as re-educating oneself and implementing practical safety measures.

Recoping involves the transformation of unhealthy coping mechanisms into healthier ones to manage stress more effectively. The goal is not merely to quit maladaptive habits but to replace them with beneficial ones. For example, quitting smoking without adopting a healthy alternative strategy to manage ongoing stress, such as from everyday work, can lead to a shift toward other unhealthy coping mechanisms like drinking, drug use, hypersexuality, gambling, or overeating. A practical illustration of recoping involves individuals who habitually suppress their emotions. These individuals can benefit from developing healthier emotional expression practices. Specific alternatives include shouting or singing in a private space like a car or vocal booth. Engaging in activities that encourage emotional expression, such as attending sporting events, joining choir vocal lessons, jamming with musical bands, taking dance classes, or participating in acting workshops, can also be beneficial. Even everyday interactions, like having regular arguments with a spouse, can provide a consistent outlet for emotional expression. However, it is important to note that recoping is a temporary measure. While it helps the body manage stress and regain balance in the short term, it does not address the underlying issues. Similar to refocusing, recoping offers immediate relief but fails to provide long-term solutions. For example, if someone lives in an area prone to frequent lightning strikes, shouting to release pent-up emotions might offer momentary relief but will not protect them from the actual threat of lightning. Without addressing the core issue, such as installing a lightning rod, this approach alone is insufficient. Recoping can help manage stress and provide temporary composure, but it must be paired with long-term strategies like re-educating oneself and implementing practical measures.

## 6. Behavioral Stress Reduction Programs (BSRPs)

Now, let us focus on the pivotal contributors in the field of stress relief, the numerous BSRPs responsible for the outcomes detailed in the extensive studies reviewed earlier. For clarity, we referred to these diverse interventions collectively as BSRPs. However, it is time to distinguish these individual approaches by name.

We begin with an overview of these programs, summarized in Table 3. This table will provide a comprehensive snapshot of each program.

Following this overview, we will delve deeper into the specifics of each BSRP, exploring their methodologies and components.

### 6.1. PsychoEducation, Paced Respiration, and Relaxation (PEPRR)

The PsychoEducation, Paced Respiration, and Relaxation (PEPRR) program incorporates psychological education, controlled breathing techniques, and relaxation exercises to reduce distress and its associated health impacts [29].

### 6.2. Animal-Assisted Therapy

Animal-Assisted Therapy (AAT) has gained popularity as an alternative approach for managing various psychological and physiological conditions. This therapeutic intervention involves interactions between patients and specially trained animals, primarily dogs, to promote emotional well-being and alleviate symptoms of stress, depression, and other health issues. Animal-Assisted Therapy utilizes the refocusing aspect of BSRPs [221,323].

### 6.3. Music Therapy

Music therapy, a clinical and evidence-based practice, utilizes music interventions to achieve personalized therapeutic goals. This practice is conducted within a therapeutic relationship by credentialed professionals who have completed approved music therapy programs. Typically, music therapists undergo either a four-year undergraduate or a two-year graduate equivalency program at an AMTA-approved institution. This rigorous training ensures that music therapists are well equipped to deliver tailored interventions that differentiate music therapy from general music listening. Music therapy aligns with refocusing, where attention is redirected to stimuli that promote relaxation [324].

### 6.4. Singing

Singing has long been associated with emotional expression and social bonding, but recent research suggests it also holds potential as a stress reduction technique. Singing encompasses both recoping and refocusing strategies [168,271].

Research on short-tailed singing mice demonstrated a link between singing behavior and stress responses. Mice displaying proactive behaviors in an open-field test, such as exploratory activity, showed higher fecal corticosterone levels but lower stress-induced corticosterone when singing. This animal study aligns with human research, where amateur choristers aged 18–85 reported an improved positive affect and social connectedness after singing. However, biological markers like cortisol and alpha-amylase did not show significant changes, suggesting that the psychological benefits of singing may be more prominent than the physiological ones [325,326].

### 6.5. Distraction

Distraction as a refocusing stress reduction technique involves diverting attention away from stress-inducing stimuli to more engaging and neutral activities. This method leverages activities such as playing games, engaging in creative tasks, or participating in physical exercises to mitigate stress responses. By focusing on alternative stimuli, individuals can experience reduced physiological and psychological stress markers. Various studies have demonstrated the effectiveness of distraction in managing stress across different settings, including medical procedures and high-pressure environments [272,300].

### 6.6. Transcendental Meditation

Transcendental Meditation (TM) is a mantra-based meditation technique designed to induce a state of “restful alertness” by focusing on the repetition of a specific word or phrase. Originating from Hinduism, TM has been adapted into a religious-neutral practice. This adaptation gained widespread popularity, partly due to its endorsement by the Beatles, who famously visited India to study Transcendental Meditation. The technique is simple, natural, and effortless, typically practiced twice daily for twenty minutes while sitting comfortably with eyes closed. The TM course, offered by the Maharishi Foundation, begins with initial lectures, followed by a personal instruction session and three days of follow-up sessions, after which individuals are encouraged to continue the practice independently [327,328,329,330].

### 6.7. Mindfulness-Based Stress Reduction and Cognitive Therapy (MBSR and MBCT)

Mindfulness practices such as Mindfulness-Based Stress Reduction (MBSR) and Mindfulness-Based Cognitive Therapy (MBCT) have roots in Buddhist traditions but have been adapted into secular, clinically standardized methods for mental health. These approaches involve focusing on the present moment with curiosity, openness, and acceptance, aiming to reduce stress and improve mental health. MBSR and MBCT primarily utilize the refocusing, recoping, and re-education aspect of BSRPs. MBSR combines mindfulness meditation, body scans, and yoga, while MBCT integrates elements of cognitive behavioral therapy (CBT) with mindfulness practices to prevent depressive relapse. Both methods have shown effectiveness in reducing anxiety and depression and improving general psychological health [114,331].

Developed by Dr. Jon Kabat-Zinn, MBSR typically involves an eight-week program with weekly group sessions, a full-day retreat, and daily homework. The techniques include mindfulness meditation, yoga, and discussions on stress and coping strategies. Participants in these programs often report reductions in perceived stress and improvements in psychological well-being. However, the evidence suggests that the benefits might vary depending on individual readiness and demographic factors such as age and psychological symptom severity [222,308,331].

### 6.8. The Teutsch IDEAL Method

The previously discussed analysis of stress and its health impacts, as well as the derived graph, reveals striking similarities to a graph proposed by Dr. Champion Kurt Teutsch in the 1960s. This graph outlines a cyclical process of stress and coping that emphasizes the importance of direct expression to achieve overall well-being and balance. The cycle, as described by Teutsch, includes the following stages:
Resentment and irritation;Suppression;Increased resentment and irritation;Indirect expression, depression, alcoholism, overeating, drug abuse, gambling, hypersexuality, compulsive buying, illness, and scapegoating.

Teutsch proposed breaking this cycle through direct expression, which he believed would lead to general well-being and equilibrium. Recent studies support the idea that expressing negative emotions can facilitate mental and physical health, while suppressing these emotions increases the susceptibility to illness. Given the common tendency to suppress negative emotions, it is crucial to recognize the long-term harm this can cause. Although recent randomized control studies on the health impacts of emotional expression among those with a habit of suppression are lacking, this remains an important area for future research [78,140,223,332,333,334,335,336,337].

The Teutsch IDEAL method, an acronym for Individualized, Directive, Explanatory, Action, Log, represents a vast and comprehensive framework that can require years of study to master fully. This method stands on three foundational pillars:
The first pillar, behavioral psychogenetics, explores the inheritance of behavior, the formation of reactions during the prenatal period and childhood, and how these can be leveraged for permanent behavioral change through re-education. This re-education aims to alter responses to previously stressful situations, impacting gene expression and inducing epigenetic changes.The second pillar, human physics, applies the principles of physics to human behavior. This approach provides a scientific framework for understanding and predicting behavior in response to various stimuli by applying strict physical laws, thereby offering a robust foundation for behavior modification.The third pillar is akin to the Mindfulness-Based Stress Reduction Program, Transcendental Meditation, and other practices rooted in Buddhism or Hinduism that employ religion-neutral, scientific methodologies derived from these religious principles to achieve consistent results. However, in the case of the IDEAL method, it religion-neutrally incorporates spiritual laws and moral principles from the Bible, interpreting them through a scientific lens.

The Teutsch IDEAL method integrates the three mechanisms of BSRP: re-education, refocusing, and recoping. The final two mechanisms are used to buy time and facilitate a deeper dive into re-education, aiming to eliminate inherited biases. The IDEAL method is designed for delivery through one-on-one sessions with a professional consultant. During these sessions, the consultant uses the method’s techniques to identify biases inherited from parents and influenced by the environment. The goal is to uncover specific misconceptions held by the client due to a lack of accurate information and to reveal the truths necessary for the client to start utilizing stimuli to their advantage, rather than misusing them and experiencing harm. By consistently achieving positive outcomes from previously harmful stimuli, the client’s stress response is mitigated, effectively neutralizing the stress-inducing nature of the situation [338,339,340,341].

### 6.9. Eye Movement Desensitization and Reprocessing (EMDR)

Eye Movement Desensitization and Reprocessing (EMDR) is a multifaceted psychotherapy that incorporates elements from various psychological approaches. EMDR’s emphasis on childhood memories aligns with psychodynamic models, while its use of trauma imagery resonates with trauma-focused CBT. Additionally, this client-centered method embraces positive and negative self-evaluations, reflecting principles of phenomenological and humanistic therapies. These diverse theoretical influences make EMDR a versatile and comprehensive therapeutic tool. EMDR primarily employs the refocusing techniques characteristic of BSRPs. Expanding on its core principles, the Professional Intervention Program for Adversity (PIPA), derived from EMDR and the Adaptive Information Processing (AIP) model, integrates low-intensity group exercises into a comprehensive group therapy framework. PIPA includes self-regulation exercises, stabilization, desensitization, reprocessing, and belief formation about the self and future [260,342].

### 6.10. Tai Chi

Tai Chi, an ancient Chinese meditative martial art, has gained popularity in the West for its potential to reduce stress and improve both physical and mental health. This practice, as an effective BSRP refocusing technique, involves gentle movements that emphasize mindfulness, structural alignment, flexibility, strength, and natural breathing. Practitioners focus on their body’s position, movements, and sensations, using imagery and rhythmic breathing to promote relaxation and dynamic stretching. The social support found in Tai Chi communities also contributes to a sense of belonging and holistic well-being, which enhances the practice’s overall impact on health [275,343].

### 6.11. Loving-Kindness and Compassion Meditation

Loving-kindness and compassion meditation (LKM and CM) are derived from Buddhist traditions and aim to cultivate an affective state of unconditional kindness and deep sympathy for those suffering. These meditative practices involve the repetition of positive phrases and the development of compassionate emotions, targeting both self-compassion and compassion for others. LKM focuses on fostering unconditional kindness, while CM aims to alleviate the suffering of all sentient beings. These methods are grounded in the belief that all living beings are interconnected, promoting emotional flexibility and psychological well-being [344,345].

### 6.12. Inquiry-Based Stress Reduction (IBSR, the Work)

Inquiry-Based Stress Reduction (IBSR), developed by Byron Katie in 1986, is a meditation technique focused on self-inquiry. It involves identifying and questioning stressful thoughts to alleviate distress. This method consists of two phases: systematically identifying stressful thoughts and investigating them through a series of four questions and turnarounds. By challenging these beliefs, participants can reduce negative emotions and adopt a more balanced approach to stress, enhancing their ability to manage distressing situations. IBSR involves refocusing and re-education techniques and integrates various psychological approaches, including psychodynamic and cognitive–behavioral models. The technique involves identifying stressful thoughts and questioning their validity through a structured inquiry. This method aims to reduce negative emotions and foster a more balanced perspective on stress-inducing situations [224,267].

### 6.13. Comprehensive Lifestyle Modification Programs (CLMPs)

The Comprehensive Lifestyle Modification Program entails a series of 10 weekly group sessions led by experienced physicians and mind–body instructors. These sessions cover a wide range of topics including yoga, stress management, mindfulness, herbal medicines, communication techniques, self-awareness, and cooking classes. Participants receive both theoretical knowledge and practical training, enabling them to integrate these practices into their daily lives. This approach aims to improve physical, psychological, and social well-being primarily through refocusing techniques within the BSRP framework [319,321].

### 6.14. Ngoma

Ngoma, rooted in Central and South African traditions, an indigenous rhythmic and dance ceremony, has been integral to healing, conflict resolution, social bonding, and spiritual experiences across various cultures for generations. The ceremony was modified to be religion-neutral and involve moderate exercise to suit a broader audience [191,346].

### 6.15. Stress Management Education

Stress management education encompasses a diverse range of strategies aimed at reducing stress and enhancing overall well-being. These methods can include recoping, refocusing, and re-education approaches. A survey of colleges of pharmacy revealed efforts to address student stress through counseling, academic advising, physical exercise support, and relationship-building activities. However, many institutions lack formal training for the faculty on student mental health and do not assess the impact of stress interventions effectively [347,348].

Adults with adverse childhood experiences often develop diverse coping strategies, ranging from problem-solving and emotion-focused approaches to maladaptive behaviors. Effective stress management education must address these vulnerabilities by promoting adaptive coping mechanisms and providing evidence-based interventions [349,350].

## 7. Conclusions

Chronic stress initiates a cascade of physiological, genetic, and psychological reactions that significantly impact health. From the initial activation of the amygdala and hypothalamus, leading to the release of catecholamines and cortisol, to the long-term effects on immune function, glucose metabolism, and neuroinflammation, the consequences of chronic stress are widespread. Chronic stress influences over 700 genes, including those involved in neurotransmission (such as serotonin and dopamine receptors), immune function, and cellular stress responses. The key genes affected include *5HTR2C*, *Abat*, *ACHE*, *ADRB*, *AP-1*, *ARHGEF1*, *Atg16l1*, *BCR-ABL*, *BDNF*, *BRCA1*, *CALM1*, *CCNI*, *CD247*, *CD28*, *CD3e*, *Cited2*, *CLDN2*, *COMT*, *Coq10b*, *CREB*, *CRHR1*, *CSF3R*, *CTNNB1*, *DAT*, *DBH*, *DPYD*, *DRD1-DRD4*, *EIF2*, *EP300*, *EPHX2*, *FKBP5*, *GAD1*, *GATA1*, *GPX1*, *GR*/*NR3C1*, *GSTP1*, *IRF3*, *NFKB/NF-kB*, *NPS*, *NPSR1*, *NPY*, *NR1H4*, *NRCAM*, *NRF2*, *NRG1*, *NRXN1*-*NRXN3*, *OXTR*, *PLCB2*, *POLM*, *PPM1F*, *PPP3CA*, *PRKCA*, *SASP*, *SCL6A4*/*5-HTT*/*5-HTTLPR*, *SIRT1*, *TP53*, *TPH2*, *USP1*, and *Wdr13*, among others. *FKBP5* is considered a critical modulator of stress responses, affecting glucocorticoid receptor activity and various cellular processes. Stress resilience, the ability to manage anxiety during stress, is influenced by genetic factors. *NPY*, *CRHR1*, and *SCL6A4* genes are linked to a lower resilience, whereas the regulation of *DRD1*-*DRD4*, *DBH*, *DAT*, and *BDNF* are associated with higher resilience.

BSRPs offer a multifaceted approach to mitigating the effects of chronic stress, including re-education, which involves changing perceptions of stressors through increased knowledge and understanding; refocusing, which redirects attention to calming stimuli such as nature, music, or mindfulness practices; and recoping, which transforms unhealthy coping mechanisms into healthier alternatives. Despite some progress in studying BSRPs, it is important to recognize that this field is still in its early stages. Ongoing research and refinement of these programs are crucial to fully harness their potential and improve health outcomes for individuals experiencing chronic stress.

## 8. Future Directions

The evolution of BSRPs necessitates significant advancements in several key areas:
Implementing more controlled randomized studies;Developing standardized methodologies;Increasing sample sizes;Extending the duration of studies to capture long-term effects;Utilizing established biomarkers for stress diagnostics;Gathering comprehensive data on biochemical changes associated with stress and disease resolution;Investigating long-term epigenetic and genetic changes influenced by BSRPs.

Reflecting on the trajectory of science and its historical tension with spirituality, it is evident that the integration of insights from various domains can foster groundbreaking advancements. In the 17th century, Galileo Galilei faced persecution for his scientific views, symbolizing the conflict between science and spirituality. Over the centuries, this dynamic has shifted, with science becoming the dominant paradigm and spirituality relegated to the background. Today, the mutual skepticism has diminished, allowing for a more integrative approach where scientific methods are applied to concepts inspired by spiritual and philosophical traditions.

Emerging fields such as music therapy, singing, distraction, and Animal-Assisted Therapy offer promising avenues for expansion and refinement within the broader context of BSRPs. These approaches, rooted in empirical research, highlight the diverse potential of integrative stress reduction techniques.

Re-education has emerged as a particularly effective long-term BSRP method, emphasizing the transformative power of knowledge and scientific literacy. By dispelling misconceptions and fostering a culture of evidence-based understanding, individuals can reduce stress and enhance their sense of control and confidence. The widespread adoption of scientific principles and habits can significantly contribute to global stress reduction.

The outlined recommendations—controlled randomized studies, standardized methodologies, larger samples, long-term measures, validated stress biomarkers, detailed biochemical data, and insights into epigenetic and genetic changes—will provide a robust foundation for BSRPs. If these strategies are effectively implemented, BSRPs could become integral to global health practices, from early education to healthcare systems.

While these projections are speculative, the path forward necessitates extensive scientific effort to transition from early adoption to mainstream acceptance. Pioneers in the field must continue their rigorous research and validation efforts, ensuring that BSRPs achieve their full potential in enhancing health and well-being worldwide.

## Figures and Tables

**Figure 1 ijerph-21-01077-f001:**
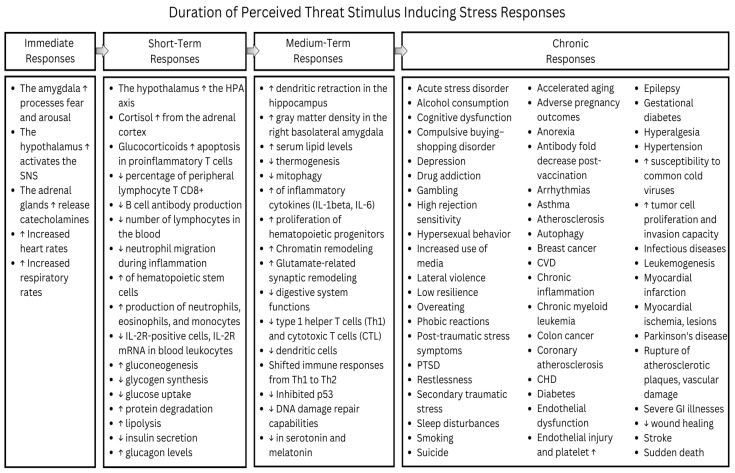
A framework illustrating the consequences based on the duration of time a stimulus is perceived as a threat, thereby inducing stress responses.

**Figure 2 ijerph-21-01077-f002:**
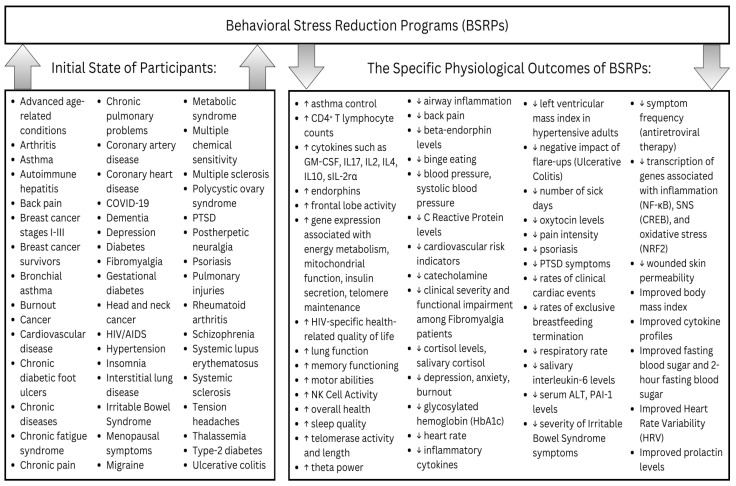
The specific physiological outcomes of BSRPs.

**Figure 3 ijerph-21-01077-f003:**
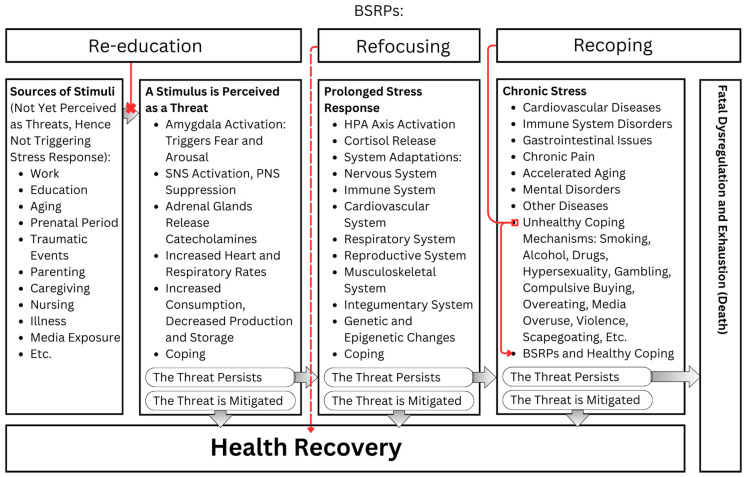
A framework illustrating how stress impacts health and the mitigating effects of BSRPs.

**Table 1 ijerph-21-01077-t001:** Genes triggered by chronic stress and their associated physiological consequences.

Gene Name	Associated with	Studies
*5HTR2C*	cardiovascular disease, depression, type-2 diabetes	[12]
*ABAT*	immune system	[13]
*ACHE*	PTSD	[14]
*ADRB*	tumor	[15]
*AP-1*	restraint stress, inflammation	[16]
*ARHGEF1*	anxiety	[17]
*ATG16L1*	immune system	[13]
*BCR-ABL*	leukemogenesis	[18,19,20]
*BDNF*	mental disorders, MSDs	[21,22,23,24]
*BRCA1*	breast cancer	[25]
*CALM1*	PTSD	[26]
*CCNI*	anxiety	[17]
*CD247*	depression	[27]
*CD28*		
*CD3e*		
*Cited2*	immune system	[13]
*CLDN2*	restraint stress, metabolism	[28]
*COMT*	neuroticism, depression	[9]
*Coq10b*	immune system	[13]
*CREB*	immune system	[16,29]
*CRHR1*	anxiety	[22,23,30]
*CSF3R*	anxiety	[17]
*CTNNB1*		
*DPYD*		
*DAT*	anxiety	[22]
*DBH*		
*DRD1*	anxiety	[22,31]
*DRD2*		
*DRD3*		
*DRD4*		
*EIF2*	PTSD	[32]
*EP300*	PTSD	[26]
*EPHX2*	depression	[33]
*FKBP5*	neuroticism, schizophrenia	[3,34]
*GAD1*	PTSD	[35]
*GATA1*	depression	[36]
*GPX1*	leukemogenesis	[20]
*GSTP1*		
*GR/NR3C1*	depression, asthma, early life and perinatal stress	[3,37,38,39]
*IRF3*	anxiety	[17]
*NFKB1/NF-kB*	immune system	[40]
*NPS*	PTSD, anxiety	[35,41]
*NPSR1*		
*NPY*	appetite	[42]
*NR1H4*	restraint stress	[28]
*NRCAM*	social defeat, abusive supervision	[43]
*NRF2*	oxidative stress	[16,32]
*NRG1*	depression	[33]
*OXTR*		
*NRXN1*	synaptic strength, depression, PTSD	[44]
*NRXN2*		
*NRXN3*		
*PLCB2*	anxiety	[17]
*POLM*		
*PPP3CA*		
*PPM1F*	PTSD, depression	[45]
*PRKCA*	PTSD	[26]
*SASP*	restraint stress	[16]
*SIRT1*	leukemogenesis	[19]
*SLC6A4/5-HTT/5-HTTLPR*	PTSD, neuroticism, anxiety, depression, early life stress, panic disorder	[31,46,47,48,49,50,51]
*TP53*	anxiety, PTSD	[17,26]
*TPH2*	mental disorders	[21]
*USP1*	ER, DNA stress	[52,53,54,55]
*WDR13*	social isolation, anhedonia, anxiety	[36]

**Table 2 ijerph-21-01077-t002:** Specific psychological and physiological outcomes of BSRPs categorized by initial unhealthy states.

State	Participants	The Specific Psychological and Physiological Outcomes of BSRPs	Critique	Study
Advanced age-related conditions	39 elderly, mean age 82 years	Improved acceptance, psychological flexibility, role limitations due to physical health; increased awareness, reduced judgment, greater self-compassion	Small sample size, no long-term follow-up, qualitative data may introduce bias	[151]
Advanced age-related conditions	40 older adults, age 55–85 years	Reduced loneliness, downregulated pro-inflammatory *NF-κB* gene expression, a trend towards reduced C Reactive Protein	Small sample size, no active control group, short duration of experiment	[77]
Advanced age-related conditions	36 Japanese older adults	Decreased salivary cortisol; correlation between decreased cortisol, negative affect, and cognitive impairment	Small sample size	[152]
Asthma	73 adults	Improved asthma control by 32%, significant reduction in exhaled nitric oxide	Small sample size, no active control group	[153]
Asthma	29 patients	Improvements in quality of life, asthma control, lung function; reduction of rescue medication, stress, enhanced psychosocial and spiritual well-being	Small sample size	[154]
Autoimmune hepatitis	17 adults, mean age 53 years	Perceived stress scores decreased by 28.6%, ALT levels decreased by 37.8%, prednisone doses reduced by 42.5%, significant decrease in IL-6, IL-8, IL-10, IL-17, IL-23	Small sample size, single-arm, non-randomized, no control group	[155]
Burnout	50 nonmedical hospital staff	Significant reductions in burnout, anxiety, and stress scores immediately after BSRP	Small sample size, no long-term follow-up, no active control group	[108]
Burnout	27 female collegiate rowers	Improved psychological well-being, decreased depression and anxiety, enhanced sleep quality, improved athletic coping skills, reduced ergometer test completion time	Small sample size, quasi-randomized design, no active control group, short follow-up duration, potential non-specific intervention effects	[156]
Burnout	90 post-pandemic era nurses	Positive emotions increased, life satisfaction improved, psychological adaptation significantly higher, negative emotions decreased, low personal accomplishment reduced, job burnout significantly lower	Small sample size, no control group, short intervention duration	[157]
Burnout	13 nursing professionals	Reduction in perceived stress, burnout, depression, and anxiety; significant increase in physical and psychological domains of quality of life	Small sample size, lack of a control group, short duration of the intervention	[158]
Burnout	37 pediatric professionals	Improvement in psychological distress, anxiety, depression, and burnout; effects maintained over 3 months	Change in hair cortisol, small sample size, retreat attendance influenced results	[159]
Burnout	88 adults, mean age 51	Immediate improvement in perceived stress, burnout, well-being, health status, back pain, and number of sick days; large effect sizes (Cohen’s d for PSQ: 1.09–1.72)	Small sample size, short duration of intervention	[160]
Burnout	53 teachers	Improvements in emotional exhaustion by 15.4% and personal accomplishment by 12.8%	Small sample size	[161]
Cancer	26 patient–caregiver dyads	Reduced cortisol and interleukin-6 levels, decreased stress and anxiety in patients	Small sample size, no control group, short intervention period	[162]
Cancer	71 patients	Significant decrease in perceived stress and post-traumatic avoidance symptoms, increase in positive states of mind and FFMQ scores	Small sample size, no long-term follow-up, no active control condition	[163]
Cancer	120 adult patients	Decreased stress and depression, increased psychological well-being	Short duration, self-reported measures	[164]
Cancer	21 couples	Significant reductions in mood disturbance, muscle tension, neurological/GI symptoms, and upper respiratory symptoms	No control group, small sample size	[165]
Cancer	268 patients, 84% women	55% reduction in mood disturbance, 29% reduction in stress symptoms, mindfulness increased significantly	No control group, improvements in some mindfulness aspects may need more time	[166]
Cancer	59 patients	Decreased cortisol levels, reduced Th1 cytokines, and lower systolic blood pressure; significant improvements in stress symptoms, quality of life, and mood	Small sample size, lack of a control group	[167]
Cancer	193 participants	Increased cytokines: GM-CSF, IL17, IL2, IL4, sIL-2rα; decreased cortisol, beta-endorphin, and oxytocin levels	No control group, results based on a single 70 min singing session	[168]
Cancer	979 adults	Medium improvements in stress, self-compassion, and social connectedness; participants practicing 5 or more days/week had greater reductions in perceived stress and increases in self-compassion	Differential study retention rates by treatment arm and self-reported engagement introduce selection bias	[169]
Cancer/Breast cancer	336 women	Reduced somatic symptoms and distress post-intervention and at 6 months; increased mindfulness at 6 months; no significant effect at 12 months	No long-term effect on somatic symptoms at 12 months, small sample size, non-active control, varied time since diagnosis	[170]
Cancer/Breast cancer	322 women	Reduced cortisol and IL-6 levels	Short-term study, no long-term data, small effect sizes, no significant group differences over 6 weeks	[171]
Cancer/Breast cancer	322 women	Significant improvement in anxiety, fear of recurrence, fatigue severity, and fatigue interference; largest effect sizes in fear of recurrence problems and fatigue severity	Small to moderate effect sizes, greater benefits observed in participants with higher baseline stress levels	[172]
Cancer/Breast cancer	36 women	Reduced blood pressure, heart rate, respiratory rate, morning cortisol, increased mindfulness state	Small sample size, no control group, short follow-up period	[173]
Cancer/Breast cancer	24 women	Significant improvement in role, cognitive, emotion, social functions, pain, fatigue, body image, future functions, therapy side effects	Small sample size, self-report tools	[174]
Cancer/Breast cancer	225 women undergoing postoperative chemotherapy	Significant improvement in physiological, social, family, emotional, and functional statuses; significant reductions in anxiety and depression	Small sample size, unclear underlying mechanisms, and lack of long-term follow-up	[175]
Cancer/Breast cancer	44 women	Anxiety decreased by 7.68, no significant changes in depression, stress, cortisol, and CRP	Small sample size, limited effect on depression, stress, and inflammatory markers	[176]
Cancer/Breast cancer	192 women newly diagnosed	Decreased stress, fatigue, sleep disturbance; increased NKCA and IFN-gamma, decreased TNF-alpha	Inconsistent practice logging	[177]
Cancer/Breast cancer	18 patients with metastasis	Mild improvements in average pain, general activity, sleep, and enjoyment of life; distress significantly reduced by 27%	Small sample size, some participants dropped out due to disease progression	[178]
Cancer/Breast cancer	82 post-treatment patients	Reduced fear of recurrence, improved physical functioning, and mediated significant reductions in perceived stress and anxiety	Small sample size, short intervention duration	[179]
Cancer/Breast cancer	101 patients (early chemotherapy)	Significant increases in post-traumatic growth, perceived social support, and quality of life	Small sample size, potential biases, and the short duration of the intervention	[180]
Cancer/Breast cancer	142 breast cancer survivors	Increased telomerase activity by 17% over 12 weeks, no significant change in telomere length	No effect on telomere length, small sample size, short study duration	[75]
Cancer/Breast cancer	15 female breast cancer survivors	Improvements in depression, state anxiety, stress, fear of recurrence, sleep quality, fatigue, and quality of life	Single-group pre–post-test design, small sample size, no control group	[181]
Cancer/Cervical cancer	22 cervical cancer survivors	Improved quality of life and modulated stress-associated biomarkers; significant increase in telomere length in CD14(+) cells and a trend in CD14(-) cells; increases in naive T-cell population correlated with telomere length in CD14(-) cells	Small sample size, no control group, short duration	[76]
Cancer/Colorectal cancer	88 patients	Higher disease control rate by 7.67%, improved functional status by 18.53%, lower depression/anxiety scores, better treatment compliance by 18.53%, improved QoL	Small sample size, no long-term follow-up, non-blinded design	[182]
Cancer/head/neck	19 patients, Australia	Reduced psychological distress and improved QoL, reductions in tension–anxiety	Small sample size, no control group, limited generalizability, no significant changes in depression–dejection	[183]
Cancer/Lung cancer	30 patients	Significant improvement in quality of life	Small sample size, short follow-up period	[184]
Cancer/Lung cancer	107 patients and partners	Patients: significant reduction in psychological distress, anxiety, depression, rumination, improved quality of life	Small sample size, high dropout, missing data, no significant effects for partners	[185]
Cardiovascular disease	74 black women	Reduced avoidance coping, slight decrease in TNF-alpha and hsCRP levels	Small sample size, no active control group, low baseline levels of psychological distress	[186]
Cardiovascular disease	60 cardiac patients	Systolic blood pressure decreased by 12%, perceived stress decreased by 45%, anger decreased by 48%	No significant effect on diastolic blood pressure, short intervention duration, no follow-up, self-reported measures	[187]
Chronic diseases	149 patients	Significant reduction in psychological distress (mean IES-R score decreased by 7.75 points), depression, and stress; improvements in psychological health sustained over 8 weeks	Small sample size, no long-term follow-up, no active control group, reliance on self-reported measures, limited gender representation	[188]
Chronic diseases	38 participants (74% white, 79% female, average age 52.6 years)	Improved mental and physical function, reduced pain, self-efficacy; decreased primary care visits by 50%, specialty care by 38%, emergency department visits by 50%, and hospital admissions by 80% at 1 year	Single-cohort design, small sample size, no control group	[189]
Chronic diseases	29 individuals	Statistically significant improvements in physical, psychological health, well-being	Small sample size, no control group	[190]
Chronic diseases	17 participants	94% found the BSRP positive, reporting increased exercise tolerance, group support, stress reduction, and beneficial spiritual experiences; none found it uncomfortable	Small sample size, no control group, short duration of the experiment, self-reported data may introduce bias	[191]
Chronic pain/Arthritis, migraine, and fibromyalgia	133 patients	Significant improvements in pain intensity, HRQoL, and psychological distress, particularly in patients with arthritis; smallest improvements in patients with chronic headache/migraine and fibromyalgia	Small sample size, no control group, varied compliance with home practice	[10]
Chronic pain/Arthritis, lower back pain, fibromyalgia	26 patients	Pain disability (22.2% improvement), psychological distress (31.1% improvement), and subjective pain rating (24.9% improvement)	Small sample size, no control group, short program length, reliance on self-reported measures, lack of follow-up data	[192]
Chronic pain/Back pain	23 patients	Improved back pain, somatic–affective depression symptoms, and frontal lobe activity	Small sample size	[193]
Chronic pain/Back pain/Low back pain	22 patients	Medium effect on health-related quality of life, psychological functioning, health-related life satisfaction, depression, and affective pain perception; pain severity improved	Small sample size, no control group, no differences in EEG power spectral density	[194]
Chronic pain/Chronic neuropathic pain	23 female breast cancer survivors	Increased posterior cingulate connectivity with medial prefrontal regions, correlated with a reduction in pain severity	Small sample size, no long-term follow-up	[195]
Chronic pain/Fibromyalgia	225 adults with fibromyalgia	BSRP superior to TAU post-treatment and at 12-month follow-up, superior to FibroQoL post-treatment, modestly better long-term in pain catastrophizing and fibromyalgianess	Small sample size, need for larger long-term studies, potential biases due to self-report measures	[196]
Chronic pain/Fibromyalgia	70 females with fibromyalgia	Reduced clinical severity, prevented IL-10 decrease, higher IL-6/IL-10 and CXCL8/IL-10 ratios associated with less improvement in psychological inflexibility	Small sample size	[197]
Chronic pain/Migraine	62 individuals, mean age 44 years	27% reduction in migraine days, significant improvements in psychological symptoms, pain self-efficacy, and sensory pain perception	Small sample size	[198]
Chronic pain/Multiple chemical sensitivity, chronic fatigue syndrome, fibromyalgia	76 women	Significant improvement in psychological distress, 5 of 9 SCL-90R subscales showed significant improvement, and 8 of 9 subscales showed significant improvement at 3-month follow-up	Control group scores unchanged, small sample size, larger sample and longer follow-up needed	[199]
Chronic pain/Postherpetic neuralgia	50 patients	Significant reduction in depression, anxiety, and pain scores	Small sample size, short intervention duration	[200]
Coronary artery disease	70 PCI patients	Decreased anxiety and depression, reduced perceived stress, increased mindfulness	Small sample size, short intervention, potential bias in telephone format	[201]
Coronary artery disease	24 patients with depression	Depressive symptoms decreased by 33%, mastery scores increased by 8.7%, improvements sustained over 12 months	Small sample size, high dropout rate, no control group, limited generalizability	[202]
Coronary artery disease	101 patients (59.4 ± 8.6 years)	Improved physical function and sum score, reduction in depression, anxiety, anger, and stress	No improvement in psychological outcomes for men, small sample size	[203]
Coronary artery disease	24 male patients	Significant improvement in psychosocial well-being; no change in basal cortisol levels or diurnal rhythm of cortisol; no correlation between stress score and cortisol levels	Small sample size, no control group; raises questions about the utility of salivary cortisol as a stress marker	[204]
Coronary heart disease	151 outpatients	Stress levels decreased significantly, anxiety decreased, distress decreased, perceived stress decreased, clinical events decreased	No significant difference in diastolic blood pressure, small sample size	[205]
Coronary heart disease	30 male patients	Significant reduction in anxiety, depression, perceived stress, systolic BP, BMI; therapeutic gains maintained at 3-month follow-up	Small sample size, only male participants, changes in medication not controlled	[206]
Coronary heart disease	56 African American patients	Myocardial flow reserve (MFR) increased by 14%, reductions in total cholesterol, LDL, triglycerides, smoking, depression, stress (GHQ)	Small sample size, limited follow-up period	[207]
COVID-19	58 asymptomatic/mild patients	Anxiety and depression scores decreased, severe anxiety and depression reduced to 0%, light anxiety and depression increased	Small sample size, no long-term effect measurement, limited technical assessment	[208]
COVID-19	84 pregnant women	Significant reductions in prenatal distress, anxiety, and childbirth fear scores	Small sample size, short-term follow-up	[209]
COVID-19	60 patients	Mental well-being scores increased by 42% in the intervention group; control group scores decreased by 6%	Small sample size, urban-only setting, short follow-up, self-reported data	[210]
Crohn’s disease	116 adults	Decreased disease activity, increased quality of life by 22%, reduced psychological symptoms by 28%, reduced fatigue by 27%	Small sample size, short duration, only mild–moderate cases	[211]
Crohn’s disease	37 patients, 18–75 years	Improved erythrocyte sedimentation rate, T-cell profiling, fecal lactoferrin, calprotectin, quality of life, emotional distress, and core self-evaluation	Small sample size	[212]
Dementia	12 family caregivers of persons with dementia	Reduced behavioral and psychological symptoms of dementia in care recipients; decreased caregiver stress, improved sleep quality	Small sample size, no control group, short intervention period	[213]
Dementia	13 middle-aged and older adults	Decreased worry and increased serenity, improved quality of life, reduced psychological distress, enhanced mindfulness and self-compassion	Small sample size, lack of control group, short follow-up period, potential bias in self-reported measures	[214]
Dementia	825 patients (11 trials)	Significant reduction in behavioral and psychological symptoms of dementia, particularly depression	No significant improvement in cognitive function, activities of daily living, agitation, or quality of life	[215]
Dementia/Frontotemporal dementia	13 individuals	Decline in anxiety (54% post-intervention, 62% at 2 months) and depression (23% at 2 months); improved coping, psychological distress, quality of life, and mindfulness skills	Small sample size, pilot study	[216]
Depression	92 students	Increased mindfulness, decreased depression, anxiety, stress, perceived stress, and anxiety sensitivity, and improved quality of life in physical health, psychological, and environmental domains	No significant improvements in social relationships, quality of life, worry, and experiential avoidance, small sample size, and potential self-selection bias among participants	[217]
Depression	78 premedical and medical students	Reduced self-reported state and trait anxiety, overall psychological distress, and depression, increased empathy	Small sample size, no long-term follow-up, potential bias, no active control group	[218]
Depression	60 elderly	Significant reduction in depression and increase in mindfulness	Small sample size, non-random sampling, limited geographic scope	[219]
Depression	60 elderly	Significant reduction in depression symptoms, improved emotion regulation and sleep quality	Small sample size; short study duration; no long-term follow-up; potential self-reporting bias	[220]
Depression	50 adults	Improvement in perceived stress, depression, psychological well-being, neuroticism, binge eating, energy, pain, and mindfulness	Small sample size, no control group	[93]
Depression	Various sample sizes across 23 studies	Significant improvements in depression, dementia, agitation, PTSD symptoms, and motor functions in multiple sclerosis and stroke	Overall low quality of studies, inconsistent methodologies	[221]
Depression	103 older adults	Clinical improvement, especially in younger, female participants with lower psychological symptom severity and those less likely to be diagnosed with depression or taking selective serotonin reuptake inhibitors	Small sample size	[222]
Depression	32 Japanese females	Improved mood states, increased salivary s-IgA levels in the depressed group; emotional inhibition had harmful effects on subjective health in the depressed group	Small sample size, short follow-up period, lack of control group	[223]
Depression	97 adults	Improvements in depression, overall psychological functioning, quality of life, anxiety, anger, and subjective happiness	Small sample size, no control group, short intervention duration	[224]
Depression	34 first-year nursing and midwifery students	Improvements in depression, anxiety, and stress scores; increased salivary immunoglobulin-A secretion; decreased salivary cortisol concentrations	Small sample size, no long-term follow-up	[225]
Diabetes/Chronic diabetic foot ulcers	8 patients, 6 family caregivers	Improved DFU healing and emotional well-being	Small sample size, qualitative study	[226]
Diabetes/Gestational diabetes	78 women	Significant reduction in fasting blood sugar, 2 h blood sugar levels, significant reduction in perceived stress	Study design was quasi-experimental	[227]
Diabetes/Type-2 diabetes	19 adults	Significant reduction in depression, lower HbA1c	Small sample size; of 15 participants, only 46.7% attended all sessions	[228]
Diabetes/Type-2 diabetes	14 adults	Decreased HA1c, arterial pressure, anxiety, depression, and general psychological distress	Small sample size, no change in body weight	[229]
Diabetes/Type-2 diabetes	56 adults	Significant reductions in serum cortisol and plasminogen activator inhibitor-1 (PAI-1) levels	Small sample size, no statistically significant differences in stress, glycemic control, vascular inflammation	[230]
Diabetes/Type-2 diabetes	48 patients	Significant improvements in fasting blood sugar and HbA1c levels	Small sample size, no long-term follow-up	[231]
HIV/AIDS	177 adults with HIV-1	Depression decreased by 20.9%, positive affect increased by 1.4%, perceived stress decreased by 7.1%, mindfulness increased by 6.3%, no significant changes in CD4+ T-cell counts, HIV-1 viral load, IL-6, CRP, or D-dimer	Psychological benefits did not sustain long-term, no significant immunological outcomes, high control group interaction	[232]
HIV/AIDS	34 adults, primarily gay	NK cell activity increased by 110%, NK cell number increased by 116%, no significant changes in psychological, endocrine, or functional health variables	Nonrandomized design, small sample size, high attrition rate, self-selected participants	[233]
HIV/AIDS	173 HIV+ patients, mean age 35.1 years	CD4 count increased up to 9 months, SCL-90R scores improved up to 6 months, MSCL scores improved up to 12 months	Randomization failure for CD4 count, small sample size, non-blinded assessments, self-reporting bias	[234]
HIV/AIDS	180 individuals aged 55+	Improvement in depression at 8 weeks (not sustained at 16 weeks), improved quality of life (QOL) at 16 weeks, no change in cognitive performance	Short-term benefits in depression and QOL, no long-term sustainment, no cognitive improvement	[235]
HIV/AIDS	22 individuals	Improved vitality, general health, physical well-being, HRQoL; reduced stress and depression; slightly lower cortisol and norepinephrine levels	No significant immune activation differences, small sample size, short follow-up	[236]
HIV/AIDS	6 HIV-positive individuals	CD4 count increased significantly and remained up to 12 months; improved SCL-90-R, but not sustained at follow-ups; no significant changes in physical status (MSCL)	Small sample size, lack of sustained psychological improvement	[237]
HIV/AIDS	76 individuals on ART	ART-related symptoms reduced by 39.1% at 3 months and 46.2% at 6 months, symptom-related distress decreased significantly	Small sample size, no significant effects on overall adherence or psychological functioning	[238]
Hypertension	38 hypertensive employees	Systolic BP reduced by 10.6 mm Hg, diastolic BP reduced by 6.3 mm Hg, decreased stress, depression, psychological distress, increased peacefulness, positive outlook, workplace satisfaction, and value of contribution	Small sample size, no long-term follow-up, no active control group, potential self-selection bias	[239]
Hypertension	872 patients (12 trials)	SBP decreased by 6.64 mmHg, DBP by 2.47 mmHg, DBP reduction sustained up to 3–6 months, no significant out-of-office BP reduction, reduced depression, anxiety, stress	Limited high-quality trials, few out-of-office BP reports, small sample sizes, lack of long-term data, inconsistent randomization	[240]
Hypertension	85 African Americans (52.8 years)	Significantly lower LVMI, significant reductions in BP and in anger	Small sample size, high attrition rate, no record of compliance for home TM practice	[241]
Hypertension	45 subjects	Systolic/diastolic blood pressure and perceived stress significantly reduced; adherence to the Mediterranean diet increased	Small sample size, no long-term follow-up	[242]
Injuries/Pulmonary	40 male veterans	Improved mental health and health-related quality of life, increased lymphocyte proliferation (PHA) and IL-17 levels	Small sample size, no active control group	[243]
Injuries/Pulmonary	40 male veterans	Improved SF-36 total score, role limitations due to physical problems, role limitations due to emotional problems, social functioning, mental health, vitality, and pain; no significant change in FEV1, FVC, or FEV1/FVC	Small sample size, no active control group	[244]
Injuries/Sport	20 university athletes	Increased pain tolerance, improved mindful awareness, notable decrease in stress/anxiety scores	Small sample size, no active control group, diverse types of injuries	[245]
Injuries/Wound	49 healthy adults	Greater mindfulness increase correlated with reduced skin permeability (days 3–4); lower IL-8 and placental growth factor levels at 22 h	Small sample size, no significant overall effect	[246]
Insomnia	111 cancer patients with insomnia	Reduced sleep onset latency, wake after sleep onset, stress and mood disturbances; increased total sleep time; improved sleep quality and dysfunctional sleep beliefs	Small sample size, no control group, need for longer follow-up	[247]
Interstitial lung disease	19 elderly patients	Improved mood and reduced stress (POMS, PSS scores); no significant changes in respiratory symptoms, lung function, or exercise tolerance	Small sample size, no control group, and unmonitored adherence to home exercises	[248]
Irritable Bowel Syndrome	24 patients	Improved quality of life and reduced IBS severity	Small sample size	[249]
Irritable Bowel Syndrome	101 patients	22.4% decrease in perceived stress, improved mental health, resilience, and quality of life, no significant impact on IBD symptoms or inflammatory biomarkers	Small sample size, no control for IBD severity indices	[250]
Menopausal symptoms	66 women, aged 47–62	Significant improvements in menopause-specific quality of life, including physical, psychosocial, and sexual dimensions	Small sample size and short follow-up duration	[251]
Menopausal symptoms	118 women	Significant improvement in menopausal symptoms, quality of life across vasomotor, psychosocial, sexual, and physical domains	Quasi-experimental design, no randomized control group	[252]
Menopausal symptoms	197 women	Reduced total menopausal symptoms at 8 months, greater reduction in anxiety and depression	Both groups improved, BSRP had more psychological benefits	[253]
Metabolic syndrome	22 individuals (11 couples)	Wives showed greater increases in physical functioning and relationship satisfaction; no significant intervention effects for husbands.	Small sample size, unequal prevalence of metabolic syndrome in couples	[254]
Multiple sclerosis	48 female patients, Iran	Significant improvement in all subscales of quality of life (QOL) post-intervention and sustained at follow-up	Small sample size, single-gender study, limited to patients with low MS severity	[255]
Multiple sclerosis	25 patients, Netherlands	Decreased depressive symptoms by 29%, increased physical QOL by 36%, increased mental QOL by 43%, reduced fatigue by 11%, enhanced visuospatial memory processing	Small sample size, high dropout rate, no control group, reliance on self-reported measures	[256]
Multiple sclerosis	48 women, Iran	Depression reduced by 46.1%, anxiety reduced by 47.2%, stress reduced by 46.6%	Small sample size, short follow-up period	[257]
Polycystic ovary syndrome	60 women	Significant reduction in mental complications, interpersonal problems, non-pregnancy physical complications, pregnancy complications, sexual complications, and religious issues	Small sample size, short follow-up, high proportion of educated participants	[258]
Psoriasis	446 participants (10 studies)	Significant improvements in physical severity (3 studies), psychological outcomes (3 studies), and quality of life (1 study); mixed results across other studies with non-significant findings	Overall low quality of evidence, heterogeneity in study design, lack of validated outcome measures in some studies, and publication bias suspected	[259]
PTSD	50 women	Decreased stress, depression, anxiety, emotion dysregulation, PTSD symptoms, increased mindfulness, lower IL-6 levels	Small sample size, no control group, short follow-up, potential selection bias	[116]
PTSD	220 individuals (war trauma)	PTSD symptoms decreased (PCL-5 scores reduced from M = 38.58 to M = 20.59), SUDS scores significantly lowered	Small sample size, no long-term follow-up, lack of quantitative measures for all segments	[260]
PTSD	30 children, adolescents	PTSD symptoms decreased by 60%, anxiety decreased by 30.5%	Small sample size, no control group, short follow-up period	[261]
PTSD	184 veterans	PTSD symptom reduction, depression reduction; those attending ≥6 sessions saw a 39.3% improvement in depression scores	Small sample size, high noncompletion rates	[262]
Rheumatoid arthritis	63 patients	At 6 months, significant improvement in psychological distress (35% reduction) and well-being, marginal improvement in depression and mindfulness	No significant changes at 2 months, small sample size, short follow-up period, no impact on RA disease activity	[263]
Schizophrenia	40 middle-aged and older patients	Improved Positive and Negative Syndrome Scale (PANSS) scores and reduced stress (DASS-stress subscale), no significant change in happiness (CHI)	Small effect size, no significant difference in happiness, limited evidence for broader applicability	[264]
Schizophrenia	137 patients	Significant increases in hope, psychological well-being, and functional recovery	Small sample size, no long-term follow-up	[265]
Schizophrenia	26 individuals	Significant decrease in perceived stress and negative affect, significant improvement in heart rate variability	No significant change in positive affect or subjective well-being, small sample size	[266]
Somatic Symptom Disorder	37 patients, mean age 37 years	Significant reductions in depression, anxiety, stress, and their severities; decreased number and severity of physical symptoms and overall SSD severity	Effective as complementary medicine, small sample size, no control group	[70]
Stuttering	56 adults who stutter	Improvements in stuttering, awareness, reactions, daily communication, quality of life, reduced anxiety, and increased life satisfaction	Small sample size, short follow-up	[267]
Stress	26 healthy individuals	Significant reduction in perceived stress; correlated with decreased gray matter density in the right basolateral amygdala	Small sample size, no control group, limited generalizability, short intervention duration	[5]
Stress	127 community residents	Decreased perceived stress, increased mindfulness and self-compassion	No control group, potential demographic bias	[8]
Stress	26 healthy practitioners and novices	Enhanced expression of genes related to energy metabolism, insulin secretion, and telomere maintenance; reduced expression of genes linked to inflammation and stress-related pathways	Small sample size, no control group, short-term study duration, need for further validation of molecular mechanisms	[60]
Stress	58 participants	Increased positivity bias in response to emotionally ambiguous signals	No control group, small sample size, short-term effects not observed, reminder issues for completing tasks promptly	[149]
Stress	84 participants	Increased likelihood of choosing reappraisal over distraction, improved sensitivity to interoceptive signals, enhanced well-being	Small sample size, lack of active control group, short duration of experiment	[268]
Stress	90 participants aged 26–55	Improved psychological well-being, hardiness, coping strategies, interpersonal interactions, and professional self-realization in adult women	Small sample size, no control group, reliance on self-reported data, potential bias in interpreting long-term parental influence	[269]
Stress	32 healthy women	Increased functional connectivity within auditory and visual networks; improved attentional focus; enhanced sensory processing and reflective awareness of sensory experience	Small sample size, no control group, short duration of the experiment	[270]
Stress	15 professional singers	Decrease in cortisol and cortisone	Increase across the high-stress condition, small sample size	[271]
Stress	77 motorcyclists, 42 +/− 14 years	Decreased N1 amplitude, relative alpha power, cortisol; increased mismatch negativity, epinephrine, heart rate, DHEA-S	Small sample size, no control group	[272]
Stress	83 individuals	Daily increases in mindfulness predicted subsequent decreases in negative affect and increases in positive affect	Magnitude of effects varied substantially between individuals, small sample size	[273]
Stress	87 adults	92% of participants strongly agreed the training was beneficial, 8% agreed	Small sample size, no control group, qualitative study	[274]
Stress	21 young adults (beginners)	Reductions in saliva cortisol, perceived mental stress at follow-up; improvements in general health perception, social functioning, vitality, and mental health/psychological well-being	Small sample size, high dropout rate, no control group	[275]
Stress	1 Buddhist monk	Significant increase in theta power, decrease in heart rate	Case study with a single participant limits generalizability	[276]
Stress/Care	29 healthcare professionals	35% reduction in distress, 30% reduction in rumination, 20% decrease in negative affect, benefits sustained over 3 months	Small sample size, no control group, short follow-up period	[277]
Stress/Care	Stress (61 parents of children with Autism)	Improvements in parental distress and parent–child dysfunctional interactions	Small sample size, no significant difference in depressive symptoms, anxiety, or life satisfaction	[278]
Stress/Care	Stress (47 parents of toddlers with developmental delays)	Decreased self-reported parenting stress and cortisol awakening response, greatest changes observed between baseline and follow-up	Small sample size, no control group	[279]
Stress/Care	Stress (100 psychiatric nurses)	Significant decreases in SCL-90, depression, anxiety, and Nursing Stress Scale scores	Short duration of study, no long-term follow-up, small sample size	[126]
Stress/Care	Stress (67 nurses in Iran)	Significant increases in mortality awareness and decreases in interpersonal problems	Quasi-experimental design, small sample size, short follow-up duration	[127]
Stress/Care	72 caregivers (COVID-19)	Anxiety decreased by 20.7%, mental component summary increased by 22.9%, physical component summary increased by 18.2%	Small sample size, no long-term follow-up	[280]
Stress/Care	64 family caregivers	Stress reduced by 8.3%, sleep quality improved by 9%, mindful awareness increased by 5.0%, no significant change in caregiver burden	Small sample size, short follow-up period, primarily self-reported outcomes	[281]
Stress/Care	155 patients and caregivers	Reduced caregiver distress	No significant difference in patient QoL	[282]
Stress/Care	148 caregivers	Reduced mental health service use	No significant differences in medical service or support group use	[283]
Stress/Care	57 family caregivers	Significant improvements in perceived stress, depressive symptoms, and subjective caregiver burden	Small sample size, no control group	[284]
Stress/Childbirth	68 pregnant women	Significantly lower perceived stress during labor and pain intensity	No significant difference in fear of childbirth, active phase of labor, second stage of delivery, total length of delivery, Apgar score, or oxytocin consumption	[285]
Stress/Childbirth	46 pregnant women	Significant increase in breastfeeding self-efficacy at 1 and 4 months postpartum, higher continuation of exclusive breastfeeding	Small sample size	[286]
Stress/Childbirth	64 breastfeeding mothers	Decreased anxiety, postpartum distress; increased breastfeeding self-efficacy	Small sample size	[287]
Stress/Education	48 students	Significant reductions in perceived stress, anger, confusion, sadness, and salivary cortisol levels.	Small sample size, no control group, potential self-reporting bias, and short duration of the intervention.	[288]
Stress/Education	119 students	Increased mindfulness, increased self-compassion	No significant reductions in trait anxiety, the quasi-experimental design and the short duration	[289]
Stress/Education	58 students and teachers	Students: reduced stress, improved self-regulation, self-efficacy, and reduced interpersonal problems; teachers: increased mindfulness, reduced interpersonal problems	Small sample size, no active control, short-term follow-up	[290]
Stress/Education	90 students	Significant improvements in mindfulness, psychological distress, perceived stress, and life satisfaction scores	Small sample size, no control group, short intervention duration, self-reported measures.	[291]
Stress/Education	302 medical students	Decreased Total Mood Disturbance by 17.8%, significant improvements in tension–anxiety, confusion–bewilderment, fatigue–inertia, and vigor–activity subscales	Initial baseline differences between groups	[292]
Stress/Education	72 nursing students	Significantly reduced stress biomarkers	Did not achieve statistically significant reductions in self-reported psychological stress during examinations	[293]
Stress/Education	49 college students	A 33% reduction in perceived stress and nearly 40% reduction in depression, anxiety, stress; increased alpha band power in frontal and occipital lobes observed in EEG, particularly during stress-inducing tasks	Small sample size, short intervention period	[294]
Stress/Media	Stress (26 middle-aged women)	Increased self-acceptance, purpose in life, relations with others, mindfulness, psychological flexibility, and well-being	Small sample size, no control group, self-selection bias, online delivery limitations	[295]
Stress/Media	131 participants	Decreased perceived stress, increased positive affect; individuals with lower baseline mindfulness exhibited greater overall improvement	Small sample size, no control group, short duration of the experiment, potential bias	[296]
Stress/Medical interventions	104 patients undergoing GI endoscopy	Reduced blood pressure, heart rate, and respiratory rate; improved perception of the procedure	Small sample size, no control group for comparison, short duration of the intervention	[297]
Stress/Medical interventions	30 patients undergoing cerebral angiography	Stable cortisol levels, reduced systolic blood pressure, decreased stress in patients with high fear levels	Small sample size, no control group, short duration of the intervention	[298]
Stress/Medical interventions	209 women undergoing TUGOR	Lower vaginal pain, higher satisfaction with pain control	No significant differences in psychological scores or stress biomarkers	[299]
Stress/Medical interventions	80 adults (dental procedures)	Significantly reduced discomfort	Small sample size, no control group, short duration	[300]
Stress/Medical interventions	100 women with breast cancer (surgical procedures)	Increased autonomy (+17.6%), environmental control (+11.8%), personal growth (+11.2%), positive relationships (+9.4%), goals in life (+12.6%), self-acceptance (+21.6%); improved sleep quality (37.0%); favorable views on oophorectomy (+14%) and mastectomy (+17%).	Small sample size	[301]
Stress/Military	49 Chinese military recruits	Improved general health, and reduced perceived stress at 12-week follow-up.	Small sample size and short follow-up duration.	[302]
Stress/Military	39 ROK Navy crew	Decreased perceived stress by 15%, anxiety by 41%; improved psychological well-being by 23%; no change in depression	Small sample size, non-equivalent control group, limited generalizability	[303]
Stress/Military	30 veterans	Reduction in perceived stress and depression; improved sleep quality; high satisfaction and compliance	Single-group design, no control group, small sample size	[304]
Stress/Trauma	63 vulnerable women	Perceived stress decreased by 20.9%	Small sample size, quasi-experimental design, short follow-up period	[305]
Stress/Trauma	24 LGB individuals	Perceived stress decreased by 23% in women and 40% in men, minority stress decreased by 12% in women	Small sample size, lack of control group, pilot study design, short follow-up period	[306]
Stress/Trauma	24 victims of gun violence	Improvements in trauma, depression, sleep quality, and life satisfaction	Uncontrolled, non-randomized trial, small sample size	[307]
Stress/Trauma	112 women with early life abuse	Increased temporal summation of heat pain intensity, reduced cortisol AUC, improved depression and emotion regulation abilities	Small sample size, no long-term follow-up, reliance on self-reported measures	[113]
Stress/Work	42 healthy working adults	Decreased perceived stress, increased mindfulness, improved sleep quality	Small sample size, use of a wait-list control group	[308]
Stress/Work	122 prison guards	Significant reduction in stress reactions, improved coping strategies, and better communication skills; in Kula prison, significant positive changes in attitudes toward detainees	Small sample size, variable significance across different prisons, no long-term follow-up, lack of control group, and potential bias in self-reported measures	[309]
Stress/Work	247 university employees	Increased job satisfaction, affective organizational commitment, perceived procedural justice, and trust in senior management; no significant change in psychological strain	Small sample size, low awareness of interventions, cross-sectional data, lack of long-term follow-up, potential response bias	[310]
Stress/Work	144 managers	Decreased perceived work-related stress (by 0.72), negative affect (by 0.93), somatic complaints intensity (by 0.69), sickness absence (by 0.69); increased self-esteem (by 1.25) and positive affect (by 0.43)	No effect on the frequency of somatic complaints, need for longer follow-up and consideration of individual differences in stress responses	[311]
Stress/Work	75 nurses and midwives	Increased sexual satisfaction	No significant effect on occupational stress and marital satisfaction, small sample size	[225]
Stress/Work	126 individuals	Improvements in perceived stress, work engagement, overall quality of life, physical health, psychological health, social relationships, and environmental	Depression and anxiety improved but did not survive Bonferroni correction; small sample size	[312]
Stroke, Dementia, Multiple Sclerosis, Spinal Cord Injury	906 participants (17 trials)	Stroke: improved gait, balance, perception of recovery, and quality of life. Dementia: reduction in stress, anxiety, sadness, depression, improved balance and cognitive function. Multiple Sclerosis: reduced spasticity, fatigue, improved balance, general health perception, and quality of life. Spinal Cord Injury: reduced effort in upper limbs, improved spasticity and mental well-being	Stroke: variability in intervention protocols. Dementia: some studies showed stable depressive symptoms and agitation. Multiple Sclerosis: mixed results in depression and constipation. Spinal Cord Injury: short-term improvements only, small sample sizes	[313]
Systemic lupus erythematosus	26 SLE patients	Improved quality of life, reduced psychological inflexibility in pain and SLE-related shame, reduced SLE symptoms, decreased depression	Small sample size, further research needed for long-term effects	[314]
Systemic lupus erythematosus	92 SLE patients	Reductions in pain, psychological dysfunction; improved perceived physical function and psychological function at 9-month follow-up	Small sample size, limited long-term maintenance of benefits	[315]
Systemic sclerosis	32 patients	Significant improvement in anger control and significant improvement in anger control and expression; no significant changes in physical functionality, anxiety, depression	Small sample size, non-random assignment, short duration, lack of significant quantitative improvements in most variables	[316]
Tension headaches	60 patients	GSI decreased by 55.2% post-intervention and 43.2% at follow-up, perceived stress decreased by 25.1% post-intervention and 20.4% at follow-up	Small sample size, limited follow-up duration	[317]
Thalassemia	66 patients	Rejection sensitivity decreased by 18.7%, resilience increased by 6.5%	Small sample size, short duration of experiment	[318]
Ulcerative colitis	47 patients	Reduced psychosocial stress, decreased physical symptoms, and improved quality of life; enhanced disease acceptance, stress management, self-confidence, and social participation	Small sample size, no control group, short experiment duration, potential self-reporting bias, and limited long-term data	[319]
Ulcerative colitis	55 adults	Reduced stress in flaring patients, prevented QOL drop during flares	Did not affect flare-up rate/severity, small sample size, short duration	[320]
Ulcerative colitis	60 patients	At 3 months: improvement in SF-36 physical function, reduction in BSI anxiety	No significant long-term effects at 12 months, small sample size	[321]
Unhealthy coping/Hypersexuality	88 bisexual men, with childhood sexual abuse	Unprotected anal insertive sex episodes reduced by 74%, depressive symptoms reduced by 43%, PTSD symptoms reduced by 56%	Small sample size, potential for bias due to self-reported measures	[85]
Unhealthy coping/Smoking	76 adult ex-smokers	Reductions in perceived stress and higher rates of smoking abstinence, reduced relapse rates	Small sample size, lack of long-term follow-up, potential self-reporting bias	[82]
Unhealthy coping/Smoking	474 individuals (4 trials)	25.2% of BSRP participants remained abstinent for more than 4 months compared to 13.6% in the usual care therapy group	Limited number of trials, need for longer follow-up periods	[322]

**Table 3 ijerph-21-01077-t003:** Behavioral Stress Reduction Programs (BSRPs).

BSRP Name	Short Description	BSRP Category
PsychoEducation, Paced Respiration, and Relaxation (PEPRR)	Integrates psychological training, regulated breathing techniques, and relaxation exercises to alleviate distress and its related health issues	Recoping, Refocusing, Re-education
Animal-Assisted Therapy	Utilizes interactions with specially trained animals, mainly dogs, to improve emotional well-being and reduce symptoms of stress and depression	Refocusing
Music Therapy	Employs structured music-based interventions delivered by certified professionals to achieve individualized therapeutic goals	Refocusing
Singing	Uses vocal expression as a means to manage emotional stress and promote psychological well-being	Recoping, Refocusing
Distraction	Shifts focus away from stressors to engaging activities such as games, creative pursuits, or physical exercise to mitigate stress responses	Refocusing
Transcendental Meditation (TM)	A mantra-based meditation technique rooted in religious-neutral Hindu traditions, involving initial guidance and independent practice for mental tranquility	Refocusing
Mindfulness-Based Stress Reduction (MBSR)	Combines mindfulness meditation, body scanning, and yoga, derived from religious-neutral Buddhist practices, to enhance present-moment awareness and reduce stress	Recoping, Refocusing, Re-education
Mindfulness-Based Cognitive Therapy (MBCT)	Blends mindfulness practices with cognitive–behavioral therapy elements, based on religious-neutral Buddhist principles, to prevent depressive relapse and manage stress	Recoping, Refocusing, Re-education
The Teutsch IDEAL Method	Encompasses behavioral psychogenetics, principles of human physics, and scientific interpretations of religious-neutral spiritual and moral teachings from the Bible	Recoping, Refocusing, Re-education
Eye Movement Desensitization and Reprocessing (EMDR)	Integrates elements of various psychological approaches, focusing on reprocessing traumatic memories and altering self-perceptions	Refocusing
Tai Chi	Involves gentle, mindful movements promoting structural alignment, flexibility, strength, and natural breathing, rooted in ancient Chinese martial arts	Refocusing
Loving-Kindness and Compassion Meditation	Focuses on cultivating unconditional kindness and empathy through the repetition of positive affirmations and compassionate thoughts, grounded in religious-neutral Buddhist traditions	Refocusing
Inquiry-Based Stress Reduction (IBSR, The Work)	Systematically challenges stressful thoughts by exploring them through a structured series of questions and cognitive reframing	Refocusing, Re-education
Comprehensive Lifestyle Modification Programs (CLMPs)	Incorporates yoga, stress management techniques, mindfulness, herbal remedies, communication skills, self-awareness, and culinary education to promote holistic well-being	Refocusing
Ngoma	Combines meditation, yoga, or prayer with rhythmic dance to traditional Congolese music, rooted in religious-neutral Central and South African cultural practices	Refocusing
Stress Management Education	Offers a wide array of strategies, including counseling, academic support, physical activity guidance, and relationship enhancement activities, to manage and reduce stress	Recoping, Refocusing

## Data Availability

No new data were created or analyzed in this study.

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
