# Peer review of "Comprehensive Review of Chronic Stress Pathways and the Efficacy of Behavioral Stress Reduction Programs (BSRPs) in Managing Diseases"

_ijerph, 2024, doi:10.3390/ijerph21081077_

Round 1

Reviewer 1 Report

Comments and Suggestions for Authors

I am grateful to the editor for the opportunity to review the manuscript of Aladdin Y. Shchaslyvyi et al. “Comprehensive Review of Chronic Stress Pathways and the Efficacy of Behavioral Stress Reduction Programs (BSRPs) in Managing Diseases.” In this review, the authors tried to consider the effectiveness of stress reduction programs on the development and progression of various diseases. The volume of this review (66 pages) and the bibliography (393 sources) indicate the authors’ claims to comprehensive coverage of this problem, which is respectable. In addition, the authors try to find confirmation for each pathogenetic (or therapeutic) factor down to the genetic level, which provides fundamental justification for the therapeutic methods under consideration.

However, I had the following comments while reviewing:

1. First of all, the scope of this review is questionable. It is very cumbersome, contains a lot of repetitions, and some of the cited publications do not directly relate to the context of the text from the review.

For example, for sections 2.2.1. Reducing: Transforming Perception Through Knowledge, 2.2.2. Refocusing: A Tool for Temporary Stress Relief, and 2.2.3. Recoping: Transitioning from Unhealthy to Healthy Coping Mechanisms The authors use a large number of references (16, 23 and 13, respectively) without attributing them to any facts.

I suggest that authors, firstly, shorten the list of cited literature by removing irrelevant references from it. Secondly, shorten the review text by removing repetitions (or, as an option, consider dividing the text into 2-3 articles).

2. The authors provide extensive information in tables 1-3 and figure 1, but do not provide links to the sources from which this information was obtained. This style of presenting information is appropriate in popular literature, but not in a scientific article. Relevant references must be provided.

3. The information provided about individual studies is too detailed, which is completely redundant in a review article. It is necessary to present this information in a more concise form.

4. Some statements are inaccurate. For example, the authors write "Early diagnosis and effective management are crucial to improving the quality of life for older adults with depression. However, the current healthcare system often falls short in providing timely and adequate mental health care for this population, exacerbating their vulnerability. For example, a 70-year-old man in a post-infarction study died suddenly while being Holter-monitored, despite no cardiac symptoms, highlighting the lethal potential of acute stress in older adults and the need for better stress management strategies [87,88 ]" (lines 370-376). A clinical example of the sudden death of a patient without symptoms does not at all indicate the need for mental health care for this population. For example, if a patient suddenly dies in his sleep, this will not indicate the need for somnologists?

Comments on the Quality of English Language

No comments

Author Response

Thank you for your patience in reviewing our extensive manuscript and for your valuable feedback. We appreciate your time, efforts, and recommendations.

We have reorganized the manuscript, repositioned several sections, and removed redundant content. For your convenience, we have marked new content in green to facilitate your review.

Comments and Responses:

Comments 1: 1. First of all, the scope of this review is questionable. It is very cumbersome, contains a lot of repetitions, and some of the cited publications do not directly relate to the context of the text from the review.

Response 1: Thank you. We have removed repetitions, resulting in fewer sections and citations.

Comments 2: For example, for sections 2.2.1. Reducing: Transforming Perception Through Knowledge, 2.2.2. Refocusing: A Tool for Temporary Stress Relief, and 2.2.3. Recoping: Transitioning from Unhealthy to Healthy Coping Mechanisms The authors use a large number of references (16, 23 and 13, respectively) without attributing them to any facts.

Response 2: We have consolidated these sections into a conclusions section. These categories emerged from the accumulated resources cited in the review.

Comments 3: I suggest that authors, firstly, shorten the list of cited literature by removing irrelevant references from it. Secondly, shorten the review text by removing repetitions (or, as an option, consider dividing the text into 2-3 articles).

Response 3: We have removed redundant BSRP studies from the main text, compiled them into a standalone table, and added two more figures, reducing the list of cited literature by 43 references and the review text by 76,466 characters. This has significantly improved the review's readability. Thank you.

Comments 4: 2. The authors provide extensive information in tables 1-3 and figure 1, but do not provide links to the sources from which this information was obtained. This style of presenting information is appropriate in popular literature, but not in a scientific article. Relevant references must be provided.

Response 4: Thank you. We have added the relevant references.

Comments 5: 3. The information provided about individual studies is too detailed, which is completely redundant in a review article. It is necessary to present this information in a more concise form.

Response 5: We have summarized this information in a concise table.

Comments 6: 4. Some statements are inaccurate. For example, the authors write "Early diagnosis and effective management are crucial to improving the quality of life for older adults with depression. However, the current healthcare system often falls short in providing timely and adequate mental health care for this population, exacerbating their vulnerability. For example, a 70-year-old man in a post-infarction study died suddenly while being Holter-monitored, despite no cardiac symptoms, highlighting the lethal potential of acute stress in older adults and the need for better stress management strategies [87,88 ]" (lines 370-376). A clinical example of the sudden death of a patient without symptoms does not at all indicate the need for mental health care for this population. For example, if a patient suddenly dies in his sleep, this will not indicate the need for somnologists?

Response 6: Thank you for pointing out this mistake. We have removed this inaccurate statement.

Thank you once again for your constructive feedback.

Reviewer 2 Report

Comments and Suggestions for Authors

Excellent review, compilation and synthesis of pertinent studies that build the science related to behavioral programs for stress reduction. The created framework was well supported.

This was an onerous read due to the length of the manuscript. I can understand wanting to include all pertinent studies and your interpretation of the findings from the review, and this will help others working in this area. However, I worry that your audience will find this interesting, but not easy to use when citing it in their work. Your reference list is extensive and will be great to review for an initial step in getting started on work in this area.

I do question, whether the manuscript should be divided so that the information is easier for reader to use. 

Author Response

Thank you for your kind words and thoughtful feedback on our review. We appreciate your time and effort in evaluating our work.

We understand your concern about the length of the manuscript. To address this, we have reorganized the content, repositioned several sections, and removed redundant material. For your convenience, we have marked the new content in green.

To improve usability, we have compiled the BSRP studies into a standalone table and added two more figures. These changes have allowed us to reduce the list of cited literature by 43 references and the review text by 76,466 characters, significantly enhancing the readability of the manuscript.

Thank you again for your valuable feedback.

Round 2

Reviewer 1 Report

Comments and Suggestions for Authors

The authors have made corrections to the text of the manuscript and responded to my comments. However, the volume of the review and the list of references are still extensive. I suggest that the authors consider dividing this review into 2 parts.

Comments on the Quality of English Language

No comments

Author Response

Dear Reviewer,

Thank you for your valuable recommendations and for striving to provide the most useful content to the readers of your journal. We share this goal wholeheartedly.

The current discourse on stress, its physiological impacts, and behavioral interventions often treats these areas as distinct, almost autonomous fields. Behavioral methods are frequently seen as supplementary, relegated to a secondary status compared to the "serious" domains of science and medicine.

One of the primary objectives of our review is to integrate these fields into a cohesive whole, demonstrating through rigorous scientific evidence how Behavioral Stress Reduction Programs (BSRPs) can lead to significant physical changes, even at the genetic level. This comprehensive approach aims to elevate the status of behavioral methods, illustrating that they are grounded in robust biological science.

Removing the extensive section on the relationship between stress and its physical consequences would undermine the central thesis and goal of our review. This foundational part is crucial for understanding the scientific basis of BSRPs. Without it, we risk perpetuating the perception that behavioral methods lack sufficient substantiation and are not integral to mainstream medical practice.

This review, in its entirety, is essential for conveying the robustness and significance of current developments in the field of behavioral stress reduction. It aims to break down outdated stereotypes that view these methods as nascent or experimental. By showcasing the depth and breadth of research, we aim to shift the prevailing mindset, demonstrating that BSRPs are credible and potent tools for addressing serious health issues.

The extensive number of studies included is vital for illustrating the substantial body of evidence supporting BSRPs. This volume is necessary to counteract existing biases and highlight the maturity and reliability of behavioral methods.

We kindly request your support in presenting this comprehensive review in its current form to help the broader scientific community recognize the substantial evidence underpinning BSRPs and their significant impact on health.